# A variational approximate posterior for the deep Wishart process

**Sebastian W. Ober**
Department of Engineering
University of Cambridge
Cambridge, UK
swo25@cam.ac.uk

**Laurence Aitchison**
Department of Computer Science
University of Bristol
Bristol, UK
laurence.aitchison@bristol.ac.uk

## Abstract

Recent work introduced deep kernel processes as an entirely kernel-based alternative to NNs (Aitchison et al. 2020). Deep kernel processes flexibly learn good top-layer representations by alternately sampling the kernel from a distribution over positive semi-definite matrices and performing nonlinear transformations. A particular deep kernel process, the deep Wishart process (DWP), is of particular interest because its prior can be made equivalent to deep Gaussian process (DGP) priors for kernels that can be expressed entirely in terms of Gram matrices. However, inference in DWPs has not yet been possible due to the lack of sufficiently flexible distributions over positive semi-definite matrices. Here, we give a novel approach to obtaining flexible distributions over positive semi-definite matrices by generalising the Bartlett decomposition of the Wishart probability density. We use this new distribution to develop an approximate posterior for the DWP that includes dependency across layers. We develop a doubly-stochastic inducing-point inference scheme for the DWP and show experimentally that inference in the DWP can improve performance over doing inference in a DGP with the equivalent prior.

## 1 Introduction

The successes of modern deep learning have highlighted that good performance on tasks such as image classification (Krizhevsky et al., 2012) requires deep models with lower layers that have the flexibility to learn good representations. Up until very recently, this was only possible in feature-based methods such as neural networks (NNs). Kernel methods did not have this flexibility because the kernel could be modified only using a few kernel hyperparameters. However, with the advent of deep kernel processes (DKPs; Aitchison et al., 2021), we now have deep kernel methods that offer neural-network like flexibility in the kernel / top-layer representation. DKPs introduce this flexibility by taking the kernel from the previous layer, then sampling from a Wishart or inverse Wishart centered on that kernel, followed by a nonlinear transformation. The sampling and nonlinear transformation steps are repeated multiple times to form a deep architecture. Remarkably, deep Gaussian processes (DGPs; Damianou & Lawrence, 2013; Salimbeni & Deisenroth, 2017), standard Bayesian NNs, infinite-width Bayesian NNs (neural network Gaussian processes or NNGPs; Lee et al., 2018; Matthews et al., 2018; Novak et al., 2019; Garriga-Alonso et al., 2019) and infinite NNs with finite width bottlenecks (Aitchison, 2020) can be written as DKPs (Aitchison et al., 2021). Indeed, for kernels that can be expressed in terms of operations on Gram matrices, Aitchison et al. (2021) showed that a particular DKP, the *deep Wishart process* (DWP) has a prior equivalent to a DGP's prior. In a DGP, the random variables inferred in variational inference are the model's intermediate features, with kernels computed as a function of these features at each layer. However, in a DWP, there are no features *at all*: the only random variables are the positive semi-definite kernel

35th Conference on Neural Information Processing Systems (NeurIPS 2021).

matrices themselves, which are sampled directly from Wishart distributions: the DWP works entirely on the kernel matrices implied by the DGP's features.

Aitchison et al. (2021) argued that DWPs should have considerable advantages over related feature-based models, because feature-based models have pervasive symmetries in the true posterior, which are difficult to capture in standard variational approximate posteriors. For instance, in a neural network, it is possible to permute rows and columns of weight matrices, such that the activations at a given layer are permuted, but the network's overall input-output function remains the same (MacKay, 1992; Sussmann, 1992; Bishop et al., 1995). These permutations result in network weights with exactly the same probability density under the true posterior, but with very different probability densities under standard variational approximate posteriors, which are generally unimodal. However, these issues do not arise with DWPs, because all permutations of the hidden units correspond to the same kernel (see Appendix D in Aitchison et al. (2021) for more details).

While Aitchison et al. (2021) showed the equivalence between DWPs and DGPs, they were not able to do inference in DWPs because they were not able to find a sufficiently flexible distribution over positive semi-definite matrices to form the basis of an approximate posterior. Instead, they were forced to work with a different DKP: the deep *inverse* Wishart processes (DIWPs), which was easier because the inverse Wishart itself forms a suitable approximate posterior. While the DIWP also avoids using features, it does not correspond directly to an already-established Bayesian model. Therefore, the DWP is a more important model, as it allows us to directly compare feature-based and kernel-based inference. In this work, we show how to create a sufficiently flexible approximate posterior for DWPs, thereby enabling us to compare directly to their equivalent DGPs. In particular, our contributions are:

- We develop a new family of flexible distributions over positive semi-definite matrices by generalising the Bartlett decomposition (Sec. 3.1).
- We use this distribution to develop an effective approximate posterior for the deep Wishart process which incorporates dependency across layers (Sec. 3.2).
- We develop a doubly stochastic inducing-point inference scheme for the DWP. While the derivation mostly follows that for deep inverse Wishart processes (Aitchison et al., 2021), we need to give a novel scheme for sampling the test/training points conditioned on the inducing points, as this is very different in the DWP compared to the previous DIWP (Sec. 3.3).
- We empirically compare DGP and DWP inference with the *exact same prior*. This was not possible in Aitchison et al. (2021) as they only derived an inference scheme for deep inverse Wishart processes, whose prior is not equivalent to a DGP prior.

We provide a reference implementation at `https://github.com/LaurenceA/bayesfunc`.

## 2 Background

### 2.1 Wishart distribution

The Wishart, $\mathcal{W}(\mathbf{\Sigma}, \nu)$, is a distribution over positive semi-definite $P \times P$ matrices, $\mathbf{G}$, with positive definite scale parameter $\mathbf{\Sigma} \in \mathbb{R}^{P \times P}$ and a positive, integer-valued degrees-of-freedom parameter, $\nu$. The Wishart distribution is defined by taking $\nu$ vectors $\mathbf{n}_\lambda \in \mathbf{R}^P$ sampled from a zero-mean Gaussian. These vectors can be generated from standard Gaussian vectors, $\boldsymbol{\xi}_\lambda$, by transforming them with the Cholesky, $\mathbf{L}$ of the scale parameter, $\mathbf{\Sigma} = \mathbf{L}\mathbf{L}^T$,

$$\mathbf{L}\boldsymbol{\xi}_\lambda = \mathbf{n}_\lambda \sim \mathcal{N}(\mathbf{0}, \mathbf{\Sigma}) \qquad \text{where } \boldsymbol{\xi}_\lambda \sim \mathcal{N}(\mathbf{0}, \mathbf{I})$$

Both $\mathbf{n}_\lambda$ and $\boldsymbol{\xi}_\lambda$ can be stacked to form $P \times \nu$ matrices, $\mathbf{N}$ and $\mathbf{\Xi}$,

$$\mathbf{N} = \begin{pmatrix} \mathbf{n}_1 & \mathbf{n}_2 & \cdots & \mathbf{n}_\nu \end{pmatrix} \qquad \mathbf{\Xi} = \begin{pmatrix} \boldsymbol{\xi}_1 & \boldsymbol{\xi}_2 & \cdots & \boldsymbol{\xi}_\nu \end{pmatrix}.$$

Wishart samples are defined by taking the sum of the outer products of the $\mathbf{n}_\lambda$'s, which can be written as a matrix-multiplication,

$$\sum_{\lambda=1}^{\nu} \mathbf{n}_\lambda \mathbf{n}_\lambda^T = \mathbf{N}\mathbf{N}^T = \mathbf{L}\mathbf{\Xi}\mathbf{\Xi}^T\mathbf{L} = \mathbf{L}\mathbf{Z}\mathbf{L}^T = \mathbf{G} \sim \mathcal{W}(\mathbf{\Sigma}, \nu) \tag{1}$$

where $\mathbf{Z} = \mathbf{\Xi}\mathbf{\Xi}^T$ is a sample from a standard Wishart (i.e. one with an identity scale parameter,)

$$\sum_{\lambda=1}^{\nu} \boldsymbol{\xi}_\lambda \boldsymbol{\xi}_\lambda^T = \mathbf{\Xi}\mathbf{\Xi}^T = \mathbf{Z} \sim \mathcal{W}\left(\mathbf{I}, \nu\right). \tag{2}$$

Note that therefore the Wishart has mean,

$$\mathbb{E}\left[\mathbf{G}\right] = \nu\,\mathbb{E}\left[\mathbf{n}_\lambda \mathbf{n}_\lambda^T\right] = \nu\mathbf{\Sigma} \tag{3}$$

## 2.2 Bartlett decomposition

However, sampling $\mathbf{\Xi}$ can be computationally expensive for very large values of $\nu$. Instead, it is possible to sample a Wishart by writing down the distribution over the Cholesky of $\mathbf{Z}$, denoted $\mathbf{A}$ (Bartlett, 1934). Taking $\mathbf{Z} = \mathbf{A}\mathbf{A}^T$, the distribution over $\mathbf{A}$ is,

$$\mathrm{P}\left(A_{jj}^2\right) = \mathrm{Gamma}\left(A_{jj}^2; \alpha{=}\tfrac{\nu-j+1}{2}, \beta{=}\tfrac{1}{2}\right), \tag{4a}$$

$$\mathrm{P}\left(A_{j>k}\right) = \mathcal{N}\left(A_{jk}; 0, 1\right). \tag{4b}$$

i.e. the square of the on-diagonal elements are Gamma distributed and the off-diagonal elements are IID standard Gaussian.

## 2.3 Deep Gaussian proceses (DGPs)

In a DGP, we progressively sample features, $\mathbf{F}_\ell$, from a Gaussian process, conditioned on features from the previous layer,

$$\mathrm{P}\left(\mathbf{F}_\ell | \mathbf{F}_{\ell-1}\right) = \prod_{\lambda=1}^{\nu_\ell} \mathcal{N}\left(\mathbf{f}_\lambda^\ell; \mathbf{0}, \mathbf{K}_{\text{features}}\left(\mathbf{F}_{\ell-1}\right)\right) \qquad \text{with } \mathbf{F}_0 = \mathbf{X}, \tag{5a}$$

$$\mathrm{P}\left(\mathbf{Y} | \mathbf{F}_{L+1}\right) = \prod_{\lambda=1}^{\nu_{L+1}} \mathcal{N}\left(\mathbf{y}_\lambda; \mathbf{f}_\lambda^{L+1}, \sigma^2\mathbf{I}\right) \tag{5b}$$

where $\mathbf{X} \in \mathbb{R}^{P \times \nu_0}$ is the input and $\mathbf{F}_\ell \in \mathbb{R}^{P \times \nu_\ell}$ are the features. We use $P$ for the number of input points and $\nu_\ell$ for the width of layer $\ell$; thus $\nu_0$ is the number of inputs and $\nu_{L+1}$ is the number of outputs. In addition, the features and targets can be written as a stack of vectors, $\mathbf{f}_\lambda^\ell \in \mathbb{R}^P$ and $\mathbf{y}_\lambda \in \mathbb{R}^P$, i.e.

$$\mathbf{F}_\ell = (\mathbf{f}_1^\ell \quad \mathbf{f}_2^\ell \quad \cdots \quad \mathbf{f}_{\nu_\ell}^\ell) \qquad\qquad \mathbf{Y} = (\mathbf{y}_1 \quad \mathbf{y}_2 \quad \cdots \quad \mathbf{y}_{\nu_{L+1}}).$$

The function $\mathbf{K}_{\text{features}}\left(\mathbf{F}_{\ell-1}\right)$ takes the features at the previous layer and returns the corresponding $P \times P$ kernel matrix. We consider isotropic kernels such as the squared exponential, which can be written as a function of $R_{ij}^{\ell-1}$, the distance between input features $i$ and $j$,

$$K_{\text{features};ij} = k(R_{ij}^{\ell-1}),$$

$$R_{ij}^{\ell-1} = \tfrac{1}{N_\ell}\sum_{\lambda=1}^{N_\ell} \left(F_{i\lambda}^{\ell-1} - F_{j\lambda}^{\ell-1}\right)^2.$$

## 2.4 Deriving equivalent deep Wishart processes

Following Aitchison et al. (2021), we show how the DGP model of Eq. (5) can be expressed as a deep Wishart process. We first consider the $P \times P$ Gram matrices defined as

$$\mathbf{G}_\ell = \tfrac{1}{\nu_\ell}\mathbf{F}_\ell\mathbf{F}_\ell^T = \tfrac{1}{\nu_\ell}\sum_{\lambda=1}^{\nu_\ell} \mathbf{f}_\lambda^\ell(\mathbf{f}_\lambda^\ell)^T,$$

where $\mathbf{f}_\lambda^\ell$ are IID and multivariate-Gaussian distributed conditioned on the features at the previous layer (Eq. 5a). Thus, $\mathbf{G}_\ell$ follows the definition of the Wishart (Eq. 1), and we can sample $\mathbf{G}_\ell$ directly,

$$\mathrm{P}\left(\mathbf{G}_\ell | \mathbf{F}_{\ell-1}\right) = \mathcal{W}\left(\mathbf{G}_\ell; \tfrac{1}{\nu_\ell}\mathbf{K}_{\text{features}}(\mathbf{F}_{\ell-1}), \nu_\ell\right).$$

To work entirely with Gram matrices rather than features, we need to be able to compute the kernel, $\mathbf{K}_{\text{features}}(\mathbf{F}_{\ell-1})$ as a function of the Gram matrix at the previous layer, $\mathbf{G}_{\ell-1}$. Remarkably, this is possible for a large family of practically relevant kernels (Aitchison et al., 2021), for instance

isotropic kernels and the arc-cosine (or ReLU) kernel. In particular, as we focus on isotropic kernels, note that it is possible to recover distances from the Gram matrix:

$$R_{ij}^\ell = \frac{1}{N_\ell} \sum_{\lambda=1}^{N_\ell} \left( \left(F_{i\lambda}^\ell\right)^2 - 2F_{i\lambda}^\ell F_{j\lambda}^\ell + \left(F_{j\lambda}^\ell\right)^2 \right) = G_{ii}^\ell - 2G_{ij}^\ell + G_{jj}^\ell.$$

Thus, since isotropic kernels depend only on the distance, it is possible to obtain $\mathbf{K}(\cdot)$, which takes the Gram matrix from the previous layer and returns the same kernel matrix as that returned by applying $\mathbf{K}_{\text{features}}$ to the features from the previous layer:

$$\mathbf{K}_{\text{features}}(\mathbf{F}_{\ell-1}) = \mathbf{K}(\mathbf{G}_{\ell-1}) = \mathbf{K}(\tfrac{1}{\nu_\ell} \mathbf{F}_{\ell-1} \mathbf{F}_{\ell-1}^T).$$

By using the equivalent kernel written as a function of the Gram matrix at the previous layer, we can entirely eliminate intermediate layer features, resulting in a deep Wishart process,

$$\mathrm{P}\left(\mathbf{G}_\ell|\mathbf{G}_{\ell-1}\right) = \mathcal{W}\left(\mathbf{G}_\ell; \tfrac{1}{\nu_\ell} \mathbf{K}(\mathbf{G}_{\ell-1}), \nu_\ell\right) \qquad \text{with } \mathbf{G}_0 = \tfrac{1}{\nu_0} \mathbf{X} \mathbf{X}^T, \tag{6a}$$

$$\mathrm{P}\left(\mathbf{F}_{L+1}|\mathbf{G}_L\right) = \prod_{\lambda=1}^{\nu_{L+1}} \mathcal{N}\left(\mathbf{f}_\lambda^{L+1}; \mathbf{0}, \mathbf{K}\left(\mathbf{G}_L\right)\right), \tag{6b}$$

$$\mathrm{P}\left(\mathbf{Y}|\mathbf{F}_{L+1}\right) = \prod_{\lambda=1}^{\nu_{L+1}} \mathcal{N}\left(\mathbf{y}_\lambda; \mathbf{f}_\lambda^{L+1}, \sigma^2 \mathbf{I}\right). \tag{6c}$$

## 2.5 The DWP formulation captures true-posterior symmetries while DGP does not

We now have two equivalent generative models: one phrased in terms of features, $\mathbf{F}_\ell$ and another phrased in terms of Gram matrices, $\mathbf{G}_\ell$. Is there any reason to prefer one over the other? It turns out that there is. In particular, consider a transformation of the features, $\mathbf{F}_\ell' = \mathbf{F}_\ell \mathbf{U}$ where $\mathbf{U}$ is a unitary matrix, such that $\mathbf{U}\mathbf{U}^T = \mathbf{I}$. Remarkably, the true posterior is symmetric under these transformations, in the sense that all unitary transformations of the underlying features have the exact same true-posterior probability density (see Aitchison et al., 2021, Appendix D.2),

$$\mathrm{P}\left(\mathbf{F}_1', \dots, \mathbf{F}_L', \mathbf{F}_{L+1}|\mathbf{X}, \mathbf{Y}\right) = \mathrm{P}\left(\mathbf{F}_1, \dots, \mathbf{F}_L, \mathbf{F}_{L+1}|\mathbf{X}, \mathbf{Y}\right).$$

It would be desirable for variational approximate posteriors to capture these true posterior symmetries. However, the usual family of Gaussian approximate posteriors over features fails to capture these symmetries because they use non-zero means. Worryingly, the failure to capture these symmetries can bias variational inference to focus on low-mass areas of the true posterior (Moore, 2016; Pourzanjani et al., 2017).

In contrast, the deep Wishart process sidesteps this issue by phrasing posteriors entirely in terms of Gram matrices, $\mathbf{G}_\ell = \frac{1}{\nu_\ell} \mathbf{F}_\ell \mathbf{F}_\ell^T$. Critically, the Gram matrix is invariant to unitary transformations of the features,

$$\mathbf{G}_\ell = \tfrac{1}{\nu_\ell} \mathbf{F}_\ell \mathbf{F}_\ell^T = \tfrac{1}{\nu_\ell} \mathbf{F}_\ell \mathbf{U}_\ell \mathbf{U}_\ell^T \mathbf{F}_\ell^T = \tfrac{1}{\nu_\ell} \mathbf{F}_\ell' \mathbf{F}_\ell'^T.$$

As such, DWP approximate posteriors written in terms of $\mathbf{G}_\ell$ implicitly respect this unitary symmetry over the features.

## 2.6 Equivalent DWP posteriors will have better ELBOs and generalisation

Let us consider the ELBOs implied by using a DGP posterior $\mathrm{Q}\left(\{\mathbf{F}_\ell\}_\ell\right)$) and the DWP posterior implied by this posterior, which for this section we denote $\mathrm{Q}\left(\{\mathbf{G}_\ell\}_\ell\right)$. Using $\mathcal{D} = (\mathbf{X}, \mathbf{Y})$:

$$\mathcal{L}_{\text{DGP}} = \mathbb{E}_\mathcal{Q}[\mathrm{P}\left(\mathbf{Y}|\mathbf{F}_{L+1}\right)] - \mathrm{KL}(\mathrm{Q}\left(\{\mathbf{F}_\ell\}_\ell\right) \| \mathrm{P}\left(\{\mathbf{F}_\ell\}_\ell|\mathcal{D}\right)),$$

$$\mathcal{L}_{\text{DWP}} = \mathbb{E}_\mathcal{Q}[\mathrm{P}\left(\mathbf{Y}|\mathbf{F}_{L+1}\right)] - \mathrm{KL}(\mathrm{Q}\left(\{\mathbf{G}_\ell\}_\ell\right) \| \mathrm{P}\left(\{\mathbf{G}_\ell\}_\ell|\mathcal{D}\right)).$$

By the data processing inequality (see e.g. Thm 6.2 in Polyanskiy & Wu (2014)), since $\mathbf{G}_\ell$ is a deterministic transformation of $\mathbf{F}_\ell$, it is straightforward to show that $\mathcal{L}_{\text{DWP}} \geq \mathcal{L}_{\text{DGP}}$. This result is one of the main motivations of the recent trend towards function-space inference (Sun et al., 2018; Ma et al., 2019); for a deeper theoretical understanding of this result we refer the reader to Burt et al. (2021), This fact can also be used to derive better PAC Bayes generalisation bounds (Section 6.1.3 of Alquier (2021)).

Note that this analysis relied on the assumption that $\mathrm{Q}\left(\{\mathbf{G}_\ell\}_\ell\right)$ would be the distribution implied by $\mathrm{Q}\left(\{\mathbf{F}_\ell\}_\ell\right)$. In practice, this is will not be the case, as we wish to directly specify our distribution in terms of Gram matrices. However, given a sufficiently flexible posterior over Gram matrices, we should still see this advantages in practice. We now turn to developing such a distribution.

# 3 Methods

As detailed in Aitchison et al. (2021), the key difficulty in obtaining a variational inference scheme for DWPs is the difficulty of providing a sufficiently flexible approximate posterior. In particular, as we are working with a probabilistic process, the number of input points, $P$, can be arbitrarily large, and thus there is always the possibility that $\nu < P$ and hence that our sampled Gram matrices are low-rank. We therefore need to form flexible variational approximate posteriors over rank $\nu$ Gram matrices. An obvious first choice is the Wishart distribution itself with degrees of freedom $\nu$, so as to match the rank of matrices sampled from the prior. However, for fixed degrees of freedom the Wishart variance,

$$\mathbb{V}[G_{ij}] = \nu \left( \Sigma_{ij}^2 + \Sigma_{ii} \Sigma_{jj} \right)$$

cannot be specified independently of the mean (Eq. 3), which is essential for a variational approximate posterior that can flexibly capture potentially narrow true posteriors. An alternative approach would be to work with a non-central Wishart, which is defined by taking $\Xi$, which is IID standard Gaussian in the case of the Wishart, to have non-zero mean. However, the non-central Wishart has a probability density function that is too difficult to evaluate in the inner loop of a deep learning algorithm (Koev & Edelman, 2006). Instead, we develop a new Generalised Singular Wishart distribution, based on the Bartlett decomposition, which modifies the Wishart to give independent control over the mean and variance of sampled matrices.

## 3.1 The Generalised Singular Wishart

To define the Generalised Singular Wishart distribution, we first need to generalise the Bartlett construction to potentially singular matrices (i.e. those for which $\nu < P$). Remembering that $\mathbf{Z} = \mathbf{A}\mathbf{A}^T$, in the singular case $\mathbf{A}$ is given by

$$\mathbf{A} = \begin{pmatrix} A_{11} & \dots & 0 \\ \vdots & \ddots & \vdots \\ A_{\nu 1} & \dots & A_{\nu\nu} \\ \vdots & \vdots & \vdots \\ A_{P1} & \dots & A_{P\nu} \end{pmatrix}, \tag{7a}$$

$$\mathrm{P}\left(A_{jj}^2\right) = \mathrm{Gamma}\left(A_{jj}^2; \tfrac{\nu-j+1}{2}, \tfrac{1}{2}\right), \qquad \mathrm{P}\left(A_{i>j}\right) = \mathcal{N}\left(A_{ij}; 0, 1\right). \tag{7b}$$

Recalling that $\mathbf{G} = \mathbf{L}\mathbf{A}\mathbf{A}^T\mathbf{L}^T$ and by applying the results of Appendices B and C, we have that

$$\mathrm{P}\left(\mathbf{G}\right) = \left( \prod_{j=1}^{P} \frac{1}{L_{jj}^{\min(j,\nu)}} \right) \prod_{j=1}^{\min(P,\nu)} \frac{\mathrm{Gamma}\left(A_{jj}^2; \frac{\nu-j+1}{2}, \frac{1}{2}\right)}{A_{jj}^{P-j} L_{jj}^{P-j+1}} \prod_{i=j+1}^{P} \mathcal{N}\left(A_{ij}; 0, 1\right).$$

In Appendix D we prove that this corresponds to the known full rank and singular Wishart distribution. Equipped with the singular Bartlett, we can now develop a generalisation of the Wishart distribution:

**Definition 1** *The Generalised Singular Wishart, $\mathcal{W}\left(\mathbf{G}; \boldsymbol{\Sigma}, \nu, \boldsymbol{\alpha}, \boldsymbol{\beta}, \boldsymbol{\mu}, \boldsymbol{\sigma}\right)$, is a distribution over positive semi-definite $P \times P$ matrices, $\mathbf{G}$, with positive definite scale matrix $\boldsymbol{\Sigma} = \mathbf{L}\mathbf{L}^T \in \mathbb{R}^{P \times P}$, a positive, integer-valued degrees-of-freedom parameter $\nu$, and Bartlett-generalising parameters $\boldsymbol{\alpha}, \boldsymbol{\beta}, \boldsymbol{\mu}, \boldsymbol{\sigma}$. These latter parameters modify the Bartlett decomposition as follows:*

$$\mathrm{Q}\left(A_{jj}^2\right) = Gamma\left(A_{jj}^2; \alpha_j, \beta_j\right) \qquad \text{for } j \leq \nu,$$
$$\mathrm{Q}\left(A_{i>j}\right) = \mathcal{N}\left(A_{ij}; \mu_{ij}, \sigma_{ij}^2\right) \qquad \text{for } j \leq \nu.$$

*This implies a distribution over $\mathbf{G} = \mathbf{L}\mathbf{A}\mathbf{A}^T\mathbf{L}^T$ with density*

$$\mathrm{Q}\left(\mathbf{G}\right) = \left( \prod_{j=1}^{P} \frac{1}{L_{jj}^{\min(j,\nu)}} \right) \prod_{j=1}^{\min(P,\nu)} \frac{1}{A_{jj}^{P-j} L_{jj}^{P-j+1}} Gamma\left(A_{jj}^2; \alpha_j, \beta_j\right) \prod_{i=j+1}^{P} \mathcal{N}\left(A_{ij}; \mu_{ij}, \sigma_{ij}^2\right).$$

Note that we use $\mathrm{Q}\left(\cdot\right)$ to reflect the fact that we will use the Generalised Singular Wishart as the basis for our approximate posterior. The density is derived using the same transformations and Jacobians as for the singular Wishart above.

## 3.2 Full approximate posterior distribution

Unlike the deep inverse Wishart process in Aitchison et al. (2021), it is not possible to obtain an optimal last-layer posterior for the deep Wishart process. Therefore, we choose a form that mimics the form of the DIWP posterior, allowing for similar across-layer dependencies:

$$\mathrm{Q}\left(\mathbf{G}_\ell|\mathbf{G}_{\ell-1}\right) = \mathcal{W}\left(\mathbf{G}_\ell; (1-q_\ell)\frac{1}{\nu_\ell}\mathbf{K}(\mathbf{G}_{\ell-1}) + q_\ell\mathbf{V}_\ell\mathbf{V}_\ell^T, \nu_\ell, \boldsymbol{\alpha}_\ell, \boldsymbol{\beta}_\ell, \boldsymbol{\mu}_\ell, \boldsymbol{\sigma}_\ell\right), \qquad (9)$$

where the approximate posterior parameters are $\{\mathbf{V}_\ell, \boldsymbol{\alpha}_\ell, \boldsymbol{\beta}_\ell, \boldsymbol{\mu}_\ell, \boldsymbol{\sigma}_\ell, q_\ell\}_{\ell=1}^L$, where $0 < q_\ell < 1$ is a scalar, and $\mathbf{V}_\ell \in \mathbb{R}^{P \times P}$. Here, the learnable parameter $q_\ell$ allows us to trade the off influence on the scale matrix from the prior and the learned covariance $\mathbf{V}_\ell\mathbf{V}_\ell^T$, which allows for similar across-layer dependencies as in (Aitchison et al., 2021). $\nu_\ell$ is fixed, as it determines the width of the layer; the remaining parameters are simply the parameters from the Bartlett generalisation.

## 3.3 Doubly stochastic inducing-point variational inference in deep inverse Wishart processes

For efficient inference in high-dimensional problems, we take inspiration from the DGP literature (Salimbeni & Deisenroth, 2017) by considering doubly-stochastic inducing-point deep Wishart processes. We begin by decomposing all variables into inducing and training (or test) points $\mathbf{X}_i \in \mathbb{R}^{P_i \times N_0}$ and $\mathbf{X}_t \in \mathbb{R}^{P_t \times N_0}$ where $P_i$ is the number of inducing points, and $P_t$ is the number of testing/training points,

$$\mathbf{X} = \begin{pmatrix} \mathbf{X}_i \\ \mathbf{X}_t \end{pmatrix}, \qquad \mathbf{F}_{L+1} = \begin{pmatrix} \mathbf{F}_i^{L+1} \\ \mathbf{F}_t^{L+1} \end{pmatrix}, \qquad \mathbf{G}_\ell = \begin{pmatrix} \mathbf{G}_{ii}^\ell & \mathbf{G}_{it}^\ell \\ \mathbf{G}_{ti}^\ell & \mathbf{G}_{tt}^\ell \end{pmatrix},$$

where e.g. $\mathbf{G}_{ii}^\ell$ is $P_i \times P_i$ and $\mathbf{G}_{it}^\ell$ is $P_i \times P_t$. The full ELBO including latent variables for all the inducing and training points is

$$\mathcal{L} = \mathbb{E}\left[\log \mathrm{P}\left(\mathbf{Y}|\mathbf{F}_{L+1}\right) + \log \frac{\mathrm{P}\left(\{\mathbf{G}_\ell\}_{\ell=1}^L, \mathbf{F}_{L+1}|\mathbf{X}\right)}{\mathrm{Q}\left(\{\mathbf{G}_\ell\}_{\ell=1}^L, \mathbf{F}_{L+1}|\mathbf{X}\right)}\right], \qquad (10)$$

where the expectation is taken over $\mathrm{Q}\left(\{\mathbf{G}_\ell\}_{\ell=1}^L, \mathbf{F}_{L+1}|\mathbf{X}\right)$. The prior is given by combining all terms in Eq. (6) for both inducing and test/train inputs,

$$\mathrm{P}\left(\{\mathbf{G}_\ell\}_{\ell=1}^L, \mathbf{F}_{L+1}|\mathbf{X}\right) = \left[\prod_{\ell=1}^L \mathrm{P}\left(\mathbf{G}_\ell|\mathbf{G}_{\ell-1}\right)\right] \mathrm{P}\left(\mathbf{F}_{L+1}|\mathbf{G}_L\right),$$

where the $\mathbf{X}$-dependence enters on the right because $\mathbf{G}_0 = \frac{1}{\nu_0}\mathbf{X}\mathbf{X}^T$. Taking inspiration from Salimbeni & Deisenroth (2017), the full approximate posterior is the product of an approximate posterior over inducing points and the conditional prior for train/test points,

$$\mathrm{Q}\left(\{\mathbf{G}_\ell\}_{\ell=1}^L, \mathbf{F}_{L+1}|\mathbf{X}\right) =$$
$$\mathrm{Q}\left(\{\mathbf{G}_{ii}^\ell\}_{\ell=1}^L, \mathbf{F}_i^{L+1}|\mathbf{X}_i\right) \mathrm{P}\left(\{\mathbf{G}_{it}^\ell\}_{\ell=1}^L, \{\mathbf{G}_{tt}^\ell\}_{\ell=1}^L, \mathbf{F}_t^{L+1}|\{\mathbf{G}_{ii}^\ell\}_{\ell=1}^L, \mathbf{F}_i^{L+1}, \mathbf{X}\right). \quad (11)$$

And the prior can be written in the same form,

$$\mathrm{P}\left(\{\mathbf{G}_\ell\}_{\ell=1}^L, \mathbf{F}_{L+1}|\mathbf{X}\right) =$$
$$\mathrm{P}\left(\{\mathbf{G}_{ii}^\ell\}_{\ell=1}^L, \mathbf{F}_i^{L+1}|\mathbf{X}_i\right) \mathrm{P}\left(\{\mathbf{G}_{it}^\ell\}_{\ell=1}^L, \{\mathbf{G}_{tt}^\ell\}_{\ell=1}^L, \mathbf{F}_t^{L+1}|\{\mathbf{G}_{ii}^\ell\}_{\ell=1}^L, \mathbf{F}_i^{L+1}, \mathbf{X}\right). \quad (12)$$

We discuss the second terms (the conditional prior) in Eq. (15). The first terms (the prior and approximate posteriors over inducing points), are given by combining terms in Eq. (6) and Eq. (9),

$$\mathrm{P}\left(\{\mathbf{G}_{ii}^\ell\}_{\ell=1}^L, \mathbf{F}_i^{L+1}|\mathbf{X}_i\right) = \left[\prod_{\ell=1}^L \mathrm{P}\left(\mathbf{G}_{ii}^\ell|\mathbf{G}_{ii}^{\ell-1}\right)\right] \mathrm{P}\left(\mathbf{F}_i^{L+1}|\mathbf{G}_{ii}^L\right), \qquad (13)$$

$$\mathrm{Q}\left(\{\mathbf{G}_{ii}^\ell\}_{\ell=1}^L, \mathbf{F}_i^{L+1}|\mathbf{X}_i\right) = \left[\prod_{\ell=2}^L \mathrm{Q}\left(\mathbf{G}_{ii}^\ell|\mathbf{G}_{ii}^{\ell-1}\right)\right] \mathrm{Q}\left(\mathbf{F}_i^{L+1}|\mathbf{G}_{ii}^L\right). \qquad (14)$$

Substituting Eqs. (11–14) into the ELBO (Eq. 10), the conditional prior cancels and we obtain,

$$\mathcal{L} = \mathbb{E}\left[\log \mathrm{P}\left(\mathbf{Y}|\mathbf{F}_t^{L+1}\right) + \log \frac{\left[\prod_{\ell=1}^L \mathrm{Q}\left(\mathbf{G}_{ii}^\ell|\mathbf{G}_{ii}^{\ell-1}\right)\right] \mathrm{Q}\left(\mathbf{F}_i^{L+1}|\mathbf{G}_{ii}^L\right)}{\left[\prod_{\ell=1}^L \mathrm{P}\left(\mathbf{G}_{ii}^\ell|\mathbf{G}_{ii}^{\ell-1}\right)\right] \mathrm{P}\left(\mathbf{F}_i^{L+1}|\mathbf{G}_{ii}^L\right)}\right].$$

The first term is a summation across test/train datapoints, and the second term depends only on the inducing points, so as in Salimbeni & Deisenroth (2017) we can compute unbiased estimates of the expectation by taking only a minibatch of datapoints. We also never need to compute the density of the conditional prior in Eq. (12), we only need to be able to sample from it,

$$\mathrm{P}\left(\{\mathbf{G}_{\mathrm{ti}}^{\ell}, \mathbf{G}_{\mathrm{tt}}^{\ell}\}_{\ell=1}^{L}, \mathbf{F}_{\mathrm{t}}^{L+1} | \{\mathbf{G}_{\mathrm{ii}}^{\ell}\}_{\ell=1}^{L}, \mathbf{F}_{\mathrm{i}}^{L+1}, \mathbf{X}\right) =$$
$$\mathrm{P}\left(\mathbf{F}_{\mathrm{t}}^{L+1} | \mathbf{F}_{\mathrm{i}}^{L+1}, \mathbf{G}_{L}\right) \prod_{\ell=1}^{L} \mathrm{P}\left(\mathbf{G}_{\mathrm{ti}}^{\ell}, \mathbf{G}_{\mathrm{tt}}^{\ell} | \mathbf{G}_{\mathrm{ii}}^{\ell}, \mathbf{G}_{\ell-1}\right). \quad (15)$$

The first distribution, $\mathrm{P}\left(\mathbf{F}_{\mathrm{t}}^{L+1} | \mathbf{F}_{\mathrm{i}}^{L+1}, \mathbf{G}_{L}\right)$, is a multivariate Gaussian, and can be evaluated using methods from the GP literature (Rasmussen & Williams, 2006; Salimbeni & Deisenroth, 2017). Specifically, we use the global inducing point scheme from Ober & Aitchison (2021). The second distribution is more difficult to sample from. To address this issue, we go back to features,

$$\mathbf{F}_{\ell}\mathbf{F}_{\ell}^{T} = \nu\mathbf{G}_{\ell} \sim \mathcal{W}\left(\mathbf{K}\left(\mathbf{G}_{\ell-1}\right), \nu\right), \quad (16)$$

with

$$\mathbf{F}_{\ell} = \begin{pmatrix} \mathbf{F}_{\mathrm{i}}^{\ell} \\ \mathbf{F}_{\mathrm{t}}^{\ell} \end{pmatrix} \qquad\qquad \mathbf{K}\left(\mathbf{G}_{\ell-1}\right) = \begin{pmatrix} \mathbf{K}_{\mathrm{ii}} & \mathbf{K}_{\mathrm{ti}}^{T} \\ \mathbf{K}_{\mathrm{ti}} & \mathbf{K}_{\mathrm{tt}} \end{pmatrix},$$

where $\mathbf{F}_{\ell} \in \mathbb{R}^{(P_{\mathrm{i}}+P_{\mathrm{t}})\times\nu_{\ell}}$, $\mathbf{F}_{\mathrm{i}} \in \mathbb{R}^{P_{\mathrm{i}}\times\nu_{\ell}}$ and $\mathbf{F}_{\mathrm{t}} \in \mathbb{R}^{P_{\mathrm{t}}\times\nu_{\ell}}$. Our goal is to sample $\mathbf{G}_{\mathrm{it}}^{\ell}$ and $\mathbf{G}_{\mathrm{tt}}^{\ell}$ given $\mathbf{G}_{\mathrm{ii}}^{\ell}$. Our approach is to note that, $\mathbf{F}_{\mathrm{t}}$ conditioned on $\mathbf{F}_{\mathrm{i}}$ is given by a matrix normal, (Eaton et al., 2007, page 310),

$$\mathrm{P}\left(\mathbf{F}_{\mathrm{t}}^{\ell} | \mathbf{F}_{\mathrm{i}}^{\ell}\right) = \mathcal{MN}\left(\mathbf{K}_{\mathrm{ti}}^{T}\mathbf{K}_{\mathrm{ii}}^{-1}\mathbf{F}_{\mathrm{i}}, \mathbf{K}_{\mathrm{tt}\cdot\mathrm{i}}, \mathbf{I}\right), \quad (17)$$

where

$$\mathbf{K}_{\mathrm{tt}\cdot\mathrm{i}} = \mathbf{K}_{\mathrm{tt}} - \mathbf{K}_{\mathrm{it}}^{T}\mathbf{K}_{\mathrm{ii}}^{-1}\mathbf{K}_{\mathrm{it}}.$$

Note that we sample each test/train point one-at-a-time/independently, in which case, $P_{\mathrm{t}} = 1$ and $\boldsymbol{\Sigma}_{22\cdot1}$ is scalar.

Then $\mathbf{G}_{\ell}$, which includes $\mathbf{G}_{\mathrm{it}}^{\ell}$ and $\mathbf{G}_{\mathrm{tt}}^{\ell}$ is given by,

$$\mathbf{G}_{\ell} = \begin{pmatrix} \mathbf{G}_{\mathrm{ii}}^{\ell} & \mathbf{G}_{\mathrm{it}}^{\ell} \\ \mathbf{G}_{\mathrm{ti}}^{\ell} & \mathbf{G}_{\mathrm{tt}}^{\ell} \end{pmatrix} = \frac{1}{\nu}\begin{pmatrix} \mathbf{F}_{\mathrm{i}}^{\ell}\left(\mathbf{F}_{\mathrm{i}}^{\ell}\right)^{T} & \mathbf{F}_{\mathrm{i}}^{\ell}\left(\mathbf{F}_{\mathrm{t}}^{\ell}\right)^{T} \\ \mathbf{F}_{\mathrm{t}}^{\ell}\left(\mathbf{F}_{\mathrm{i}}^{\ell}\right)^{T} & \mathbf{F}_{\mathrm{t}}^{\ell}\left(\mathbf{F}_{\mathrm{t}}^{\ell}\right)^{T} \end{pmatrix} = \frac{1}{\nu}\mathbf{F}_{\ell}\mathbf{F}_{\ell}^{T}$$

For $\mathbf{F}_{\mathrm{i}}$, we can use any value as long as $\mathbf{G}_{\mathrm{ii}}^{\ell} = \mathbf{F}_{\mathrm{i}}^{\ell}\left(\mathbf{F}_{\mathrm{i}}^{\ell}\right)^{T}$, as the resulting distribution over $\mathbf{G}_{\ell}$ arising from Eq. (16) does not depend on the specific choice of $\mathbf{F}_{\mathrm{i}}$ (App. E). Remembering that to sample $\mathbf{G}_{\mathrm{ii}}$, we explicitly sample its potentially low-rank Cholesky, $\mathbf{L}_{\ell}\mathbf{A}_{\ell}$, we can directly use

$$\mathbf{F}_{\mathrm{i}}^{\ell} = \mathbf{L}_{\ell}\mathbf{A}_{\ell}$$

However, this only works if $\nu \leq P_{\mathrm{i}}$, in which case, $\mathbf{L}_{\ell}\mathbf{A}_{\ell} \in \mathbb{R}^{P_{\mathrm{i}}\times\nu}$. In the unusual case where we have fewer inducing points than degrees of freedom, $P_{\mathrm{i}} < \nu$, then $\mathbf{L}_{\ell}\mathbf{A}_{\ell} \in \mathbb{R}^{P_{\mathrm{i}}\times P_{\mathrm{i}}}$, so we need to pad to achieve the required size of $P_{\mathrm{i}} \times \nu_{\ell}$,

$$\mathbf{F}_{\mathrm{i}}^{\ell} = \begin{pmatrix} \mathbf{L}_{\ell}\mathbf{A}_{\ell} & \mathbf{0} \end{pmatrix}.$$

Finally, note that we can optimise all the variational parameters using standard reparameterised variational inference (Kingma & Welling, 2014; Rezende et al., 2014). For an algorithm, see Alg. 1.

## 3.4 Computational complexity

Recalling that $\nu_{\ell}$ is the width of the $\ell$th layer, $P_i$ is the number of inducing points, and $P_t$ is the number of train or test points, the computational complexity of one DWP layer is given by $O(P_i^3 + P_t P_i^2)$. This is a decrease of a factor of $\nu_{\ell+1}$ over the complexity for standard DGP inference, such as doubly stochastic variational inference (Salimbeni & Deisenroth, 2017), which has complexity $O(\nu_{\ell+1}(P_i^3 + P_t P_i^2))$. The difference arises from the fact that in a DGP, $\nu_{\ell+1}$ Gaussian processes are sampled in each layer, whereas for a DWP we sample a single Gram matrix.

---

**Algorithm 1** Computing predictions/ELBO for one batch

---

**P parameters:** $\{\nu_\ell\}_{\ell=1}^L$.
**Q parameters:** $\{\mathbf{V}_\ell, q_\ell, \boldsymbol{\alpha}_\ell, \boldsymbol{\beta}_\ell, \boldsymbol{\mu}_\ell, \boldsymbol{\sigma}_\ell\}_{\ell=1}^L, \mathbf{X}_i$.
**Inputs:** $\mathbf{X}_t$; **Targets:** $\mathbf{Y}$

combine inducing and test/train inputs
$\mathbf{X} = (\mathbf{X}_i \quad \mathbf{X}_t)$
sample first Gram matrix and update ELBO
$\mathbf{G}_0 = \frac{1}{\nu_0} \mathbf{X}\mathbf{X}^T$
**for** $\ell$ **in** $\{1, \ldots, L\}$ **do**

    sample inducing Gram matrix and its Cholesky, $\mathbf{L}_\ell \mathbf{A}_\ell$ and update ELBO
    $\mathbf{L}_\ell \mathbf{A}_\ell \mathbf{A}_\ell^T \mathbf{L}_\ell^T = \mathbf{G}_{ii}^\ell \sim Q\left(\mathbf{G}_{ii}^\ell | \mathbf{G}_{ii}^{\ell-1}\right)$
    $\mathcal{L} \leftarrow \mathcal{L} + \log P\left(\mathbf{G}_{ii}^\ell | \mathbf{G}_{ii}^{\ell-1}\right) - \log Q\left(\mathbf{G}_{ii}^\ell | \mathbf{G}_{ii}^{\ell-1}\right)$
    sample full Gram matrix from conditional prior
    $\boldsymbol{\Sigma} = \frac{1}{\nu_\ell} \mathbf{K}(\mathbf{G}_{\ell-1})$
    $\boldsymbol{\Sigma}_{tt \cdot i} = \boldsymbol{\Sigma}_{tt} - \boldsymbol{\Sigma}_{it}^T \boldsymbol{\Sigma}_{ii}^{-1} \boldsymbol{\Sigma}_{it}$
    $\mathbf{F}_i^\ell = \mathbf{L}_\ell \mathbf{A}_\ell$
    $\mathbf{F}_t^\ell \sim \mathcal{MN}\left(\boldsymbol{\Sigma}_{ti}^T \boldsymbol{\Sigma}_{ii}^{-1} \hat{\mathbf{F}}_i, \boldsymbol{\Sigma}_{tt \cdot i}, \mathbf{I}\right)$
    $\mathbf{G}_\ell = \begin{pmatrix} \mathbf{G}_{ii}^\ell & \hat{\mathbf{F}}_i^\ell (\hat{\mathbf{F}}_t^\ell)^T \\ \hat{\mathbf{F}}_t^\ell (\hat{\mathbf{F}}_i^\ell)^T & \hat{\mathbf{F}}_t^\ell (\hat{\mathbf{F}}_t^\ell)^T \end{pmatrix}$
**end for**
sample GP inducing outputs and update ELBO
$\mathbf{F}_i^{L+1} \sim Q\left(\mathbf{F}_i^{L+1} | \mathbf{G}_{ii}^L\right)$
$\mathcal{L} \leftarrow \mathcal{L} + \log P\left(\mathbf{F}_i^{L+1} | \mathbf{G}_{ii}^L\right) - \log Q\left(\mathbf{F}_i^{L+1} | \mathbf{G}_{ii}^L\right)$
sample GP predictions conditioned on inducing points
$\mathbf{F}_t^{L+1} \sim Q\left(\mathbf{F}_t^{L+1} | \mathbf{G}^L, \mathbf{F}_i^{L+1}\right)$
add likelihood to ELBO
$\mathcal{L} \leftarrow \mathcal{L} + \log P\left(\mathbf{Y} | \mathbf{F}_t^{L+1}\right)$

---

## 4 Results

The DWP prior is equivalent to a DGP prior (Sec. 2.4) (Aitchison et al., 2021); the only difference is that in a DGP, we use features as the latent variables, whereas in the DWP we use Gram matrices. Using Gram matrices as in the DWP should be beneficial as the true posteriors are expected to be simpler than in the DGP (Sec. 2.5). We perform a comparison of the posterior intermediate-layer features implied by a DWP and those of a DGP on a toy problem in App. G; these show the benefits of being more able to model the true posterior.

For more quantitative experiments, we trained a DWP and a DGP with the exact same generative model with a squared exponential kernel on the UCI datasets from Gal & Ghahramani (2015). We trained both models for 20000 gradient steps using the Adam optimizer Kingma & Ba (2015); we detail the exact experimental setup in Appendix F. We report ELBOs and test log likelihoods for depth 5 in Table 1; we report other depths and quote the relevant results from Aitchison et al. (2021) for reference in Appendix H. We found that the DWP often outperformed the DGP model, especially evident if we look at the ELBOs and smaller datasets (boston, concrete, energy, and yacht). On larger datasets, the benefits often disappear, as accurate uncertainty modelling is less relevant. Note that we compared against the recently introduced DGP method based on global inducing points (Ober & Aitchison, 2021). Global inducing point methods were particularly important in our setting because we use a standard feedforward architecture without skip connections to ensure equivalence between the DGP and DWP. Standard DSVI has considerable difficulties with optimizing the approximate posterior in such models; to get optimization to work effectively Salimbeni & Deisenroth (2017) were forced to modify the prior to introduce skip connections.

Table 1: ELBOs and log-likelihoods for UCI datasets from (Gal & Ghahramani, 2015) for a five-layer network. See Appendix H for other depths. Significantly better results are highlighted.

| | dataset | DWP | DGP |
|---|---|---|---|
| ELBO | boston | **-0.38 ± 0.01** | -0.45 ± 0.01 |
| | concrete | **-0.49 ± 0.00** | -0.50 ± 0.00 |
| | energy | **1.41 ± 0.00** | 1.39 ± 0.00 |
| | kin8nm | -0.14 ± 0.00 | **-0.13 ± 0.00** |
| | naval | 3.65 ± 0.07 | **3.91 ± 0.10** |
| | power | **0.03 ± 0.00** | 0.02 ± 0.00 |
| | protein | -1.01 ± 0.00 | **-1.00 ± 0.00** |
| | wine | -1.19 ± 0.00 | -1.19 ± 0.00 |
| | yacht | **1.65 ± 0.01** | 1.37 ± 0.03 |
| LL | boston | -2.39 ± 0.04 | -2.44 ± 0.04 |
| | concrete | -3.13 ± 0.01 | -3.14 ± 0.02 |
| | energy | -0.70 ± 0.03 | -0.70 ± 0.03 |
| | kin8nm | **1.40 ± 0.00** | 1.38 ± 0.01 |
| | naval | 8.21 ± 0.05 | 8.31 ± 0.07 |
| | power | -2.77 ± 0.01 | -2.78 ± 0.01 |
| | protein | -2.73 ± 0.00 | -2.73 ± 0.00 |
| | wine | -0.96 ± 0.01 | -0.96 ± 0.01 |
| | yacht | **-0.43 ± 0.10** | -0.88 ± 0.08 |

## 4.1 Runtimes & training curves

In Sec. 3.4, we showed that DWPs have a lower computational complexity than DGPs, because of the need for DGPs to sample $\nu_\ell$ features in each layer, whereas DWPs only need to sample one Gram matrix. Here, we briefly discuss the runtimes of our implementations. We show a plot of the training curves for one split of boston with a 5-layer DGP and DWP in Fig. 1, plotted against both runtime and epoch. From these plots, we make two observations. First, the DWP trains much more quickly than the DGP in terms of runtime. However, it seems to require slightly more epochs than the DGP to converge (note that the spike at the start of the DGP curve is an artifact of the tempering scheme we use). In Appendix H, we provide a table of time per epoch, which shows that we obtain faster runtime for protein and for shallower models, although the gains are slightly more modest due to the models being shallower and the fact that we run protein on a GPU, as opposed to a CPU for boston.

## 5 Related Work

The DWP prior was introduced by Aitchison et al. (2021). However, they were not able to do variational inference with the DWP because they did not have a sufficiently flexible approximate posterior over positive semi-definite matrices. Instead, they were forced to work with a deep *inverse* Wishart process, which is easier because the inverse Wishart itself is a suitable approximate posterior. Here, we give a flexible generalised Wishart distribution over positive semi-definite matrices which is suitable for use as a variational approximate posterior in the DWP. As the deep Wishart process prior is equivalent to a DGP prior, we were able to directly compare DGP and DWP inference in models with the exact same prior. Such a comparison with equivalent priors was not possible in Aitchison et al. (2021), because their deep *inverse* Wishart process priors are not equivalent to DGP priors.

There is an alternative line of work using *generalised* Wishart processes (Wilson & Ghahramani, 2011, as opposed to our *deep* Wishart processes). Note that the "generalised Wishart process" terminology does not seem to have spread as widely as it should, but it is very useful in our context. A generalised Wishart process specifies a distribution over an infinite number of finite-dimensional Wishart-distributed matrices. These matrices might represent e.g. the noise covariance in a dynamical system, in which case there might be an infinite number of such matrices, one for each time or location in the state-space (Wilson & Ghahramani, 2011; Heaukulani & van der Wilk, 2019; Jorgensen et al., 2020). In contrast, the Wishart process (Dawid, 1981; Bru, 1991) describes finite dimensional marginals of a single, potentially infinite dimensional matrix. In our context, we stack

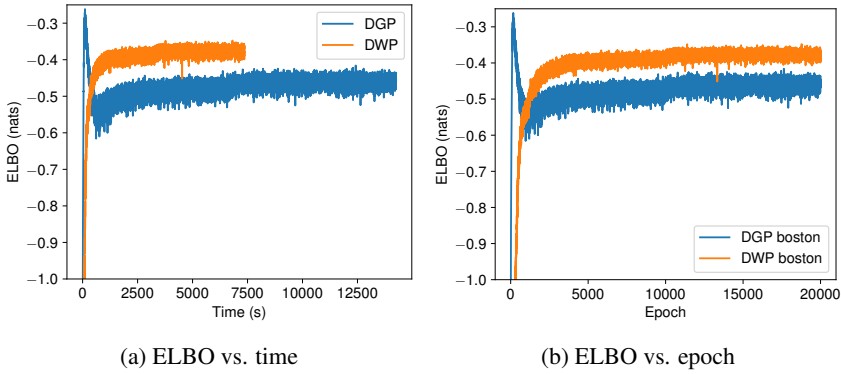

(a) ELBO vs. time            (b) ELBO vs. epoch

Figure 1: ELBO versus time and epoch for five-layer models on one split of boston

(non-generalised) Wishart processes to form a deep Wishart process. Importantly, these generalised Wishart priors do not have the flexibility to capture a DGP prior because the underlying features at all locations are jointly multivariate Gaussian (Sec. 4 in Wilson & Ghahramani, 2011) and therefore lack the required nonlinearities between layers. Further, not only do the underlying stochastic processes (deep vs generalised Wishart process) differ, inference is also radically different. In particular, work on the generalised Wishart performs inference on the underlying multivariate Gaussian feature vectors (Eq. 1 e.g. Eq. 15-18 in Wilson & Ghahramani 2011, Eq. 12 in Heaukulani & van der Wilk 2019 or Eq. 24 in Jorgensen et al. 2020). Unfortunately, variational approximate posteriors defined over multivariate Gaussian feature vectors fail to capture symmetries in the true posterior (Sec. 2.5). In contrast, we define approximate posteriors directly over the symmetric positive semi-definite Gram matrices themselves, which required us to develop new, more flexible distributions over these matrices.

## 6 Limitations

There are a few limitations of our work. First, it is only possible to derive equivalent DWPs for certain kernels - namely, those where we can skip the feature representation and work entirely in Gram matrices. While this holds for a large range of kernels, such as isotropic kernels and the arc-cosine kernel, it does not hold for some common kernels such as automatic relevance determination (ARD) kernels. However, we note that we are able to use any kernel for the first layer, and that in practice we did not find that ARD kernels in intermediate layers significantly improved performance, as all the features have a shared prior. The second limitation is that it is not currently possible to incorporate modifications to the basic DGP model such as skip connections (Duvenaud et al., 2014; Salimbeni & Deisenroth, 2017). Such modifications would require the use of the non-central Wishart distribution, which is difficult to evaluate in the inner loop of a deep learning algorithm (Koev & Edelman, 2006). Third, the generalisation of our work to more complex architectures such as convolutional models is non-trivial, and will be left to future work. Finally, it seems that the performance is not as competitive for larger datasets, where uncertainty representation is of more limited use - we will explore improving this in future work.

## 7 Conclusions

We introduced a flexible distribution over positive semi-definite matrices which formed the basis of a variational approximate posterior for the deep Wishart process. We adapted the doubly stochastic variational inference scheme from Aitchison et al. (2021) to the deep Wishart process. Thus, we were able to directly compare the performance for inference in a DWP vs. DGP with exactly the same prior. This isolates the effects on performance of the prior and the inference procedure. We found DWPs often have improved performance over their corresponding equivalent DGP models, particularly in the ELBOs, indicating improved uncertainty representation.

There are no anticipated social impacts as the work is largely theoretical.

## Acknowledgements

The authors thank David R. Burt, Mark van der Wilk, Samuel Power, and Adam X. Yang for helpful discussions, and would like to thank the reviewers for their comments.

## Funding Transparency

SWO acknowledges funding from the Gates Cambridge Trust for his doctoral studies. The authors have no competing interests or additional funding to declare.

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
