# A Deriving Jacobians for matrix transformations

Following the approach in (Mathai, 1997; Mathai & Haubold, 2008), we define the Jacobian of a function from $x$ to $y$ as the ratio of volume elements,

$$\text{jacobian} = \frac{dy_1 dy_2 \cdots dy_N}{dx_1 dx_2 \cdots dx_N}.$$

Critically, $dx_i$ and $dy_i$ are basis-vectors, *not* scalars. As we are multiplying vectors, not scalars, we need to be careful about our choice of multiplication operation. The correct choice in our context is an anti-symmetric exterior product, representing a directed area or volume element, such that,

$$dx_i dx_j = -dx_j dx_i.$$

As the product is antisymmetric, the product of a basis-vector with itself is zero,

$$dx_i dx_i = -dx_i dx_i = 0,$$

which makes sense because the product represents an area, and the area is zero if the two vectors are aligned. To confirm that this matches usual expressions for Jacobians, consider a $2 \times 2$ matrix-vector multiplication, $\mathbf{y} = \mathbf{Ax}$:

$$\begin{pmatrix} dy_1 \\ dy_2 \end{pmatrix} = \begin{pmatrix} A_{11} & A_{12} \\ A_{21} & A_{22} \end{pmatrix} \begin{pmatrix} dx_1 \\ dx_2 \end{pmatrix} = \begin{pmatrix} A_{11} dx_1 + A_{12} dx_2 \\ A_{21} dx_1 + A_{22} dx_2 \end{pmatrix}.$$

Thus,

$$dy_1 dy_2 = (A_{11} dx_1 + A_{12} dx_2)(A_{21} dx_1 + A_{22} dx_2)$$
$$dy_1 dy_2 = A_{11} A_{21} dx_1^2 + A_{11} A_{22} dx_1 dx_2 + A_{12} A_{21} dx_2 dx_1 + A_{12} A_{22} dx_2^2$$

As $dx_1^2 = dx_2^2 = 0$, and $dx_1 dx_2 = -dx_1 dx_2$, we have,

$$dy_1 dy_2 = (A_{11} A_{22} - A_{12} A_{21}) dx_1 dx_2$$
$$dy_1 dy_2 = |\mathbf{A}| dx_1 dx_2,$$

so that the Jacobian computed using the determinant definition is equivalent to the expression for the determinant obtained by working with volume elements.

# B Jacobian for the product of a lower triangular matrix with itself

In this section, we compute the Jacobian for the transformation from $\mathbf{\Lambda} = \mathbf{LA}$ to $\mathbf{G} = \mathbf{\Lambda\Lambda}^T$. We begin by noting that the top-left block of the product of a lower-triangular matrix with itself is a product of smaller lower-triangular matrices:

$$\begin{pmatrix} \mathbf{\Lambda}_{:N,:N} & \mathbf{0} \\ \mathbf{\Lambda}_{N:,:N} & \mathbf{\Lambda}_{:N,:N} \end{pmatrix} \begin{pmatrix} \mathbf{\Lambda}_{:N,:N}^T & \mathbf{\Lambda}_{N:,:N}^T \\ \mathbf{0} & \mathbf{\Lambda}_{:N,:N}^T \end{pmatrix} = \begin{pmatrix} \mathbf{\Lambda}_{:N,:N} \mathbf{\Lambda}_{:N,:N}^T & \cdots \\ \vdots & \ddots \end{pmatrix}.$$

As such, we can incrementally compute the Jacobian for this transformation by starting with the top-left $1 \times 1$ matrix,

$$G_{11} = \Lambda_{11}^2 \tag{18}$$
$$dG_{11} = 2\Lambda_{11} d\Lambda_{11}. \tag{19}$$

Next, we consider the top-left $2 \times 2$ matrix,

$$\begin{pmatrix} G_{11} & G_{12} \\ G_{21} & G_{22} \end{pmatrix} = \begin{pmatrix} \Lambda_{11} & 0 \\ \Lambda_{21} & \Lambda_{22} \end{pmatrix} \begin{pmatrix} \Lambda_{11} & \Lambda_{21} \\ 0 & \Lambda_{22} \end{pmatrix} = \begin{pmatrix} \Lambda_{11}^2 & \Lambda_{21}\Lambda_{11} \\ \Lambda_{21}\Lambda_{11} & \Lambda_{21}^2 + \Lambda_{22}^2 \end{pmatrix}.$$

Thus

$$dG_{21} = \Lambda_{21} d\Lambda_{11} + \Lambda_{11} d\Lambda_{21}$$
$$dG_{22} = 2\Lambda_{22} d\Lambda_{22} + 2\Lambda_{21} d\Lambda_{21}.$$

Combining $dG_{11}$ and $dG_{21}$ gives

$$dG_{11}dG_{21} = (2\Lambda_{11}d\Lambda_{11})(\Lambda_{21}d\Lambda_{11} + \Lambda_{11}d\Lambda_{21})$$
$$dG_{11}dG_{21} = 2\Lambda_{11}^2 (d\Lambda_{11}d\Lambda_{21}),$$

and then combining $dG_{11}dG_{21}$ and $dG_{22}$ gives

$$dG_{11}dG_{21}dG_{22} = \left(2\Lambda_{11}^2 (d\Lambda_{11}d\Lambda_{21})\right)(2\Lambda_{22}d\Lambda_{22} + 2\Lambda_{21}d\Lambda_{21})$$
$$dG_{11}dG_{21}dG_{22} = 4\Lambda_{11}^2\Lambda_{22}(d\Lambda_{11}d\Lambda_{21}d\Lambda_{22}).$$

Thus, we can prove by induction that the volume element for the top-left $p \times p$ block of $\mathbf{G}$, and in addition the first $K < p+1$ off-diagonal elements of the $p+1$th row is

$$\underbrace{\left(\prod_{i=1}^{p}\prod_{k=1}^{i} dG_{ik}\right)}_{\text{vol. elem. for } \mathbf{G}_{:p,:p}} \underbrace{\left(\prod_{k=1}^{K} dG_{p+1,k}\right)}_{\text{vol. elem. for } \mathbf{G}_{p+1,:K}} = 2^p \left(\prod_{i=1}^{p}\prod_{k=1}^{i} \Lambda_{kk}d\Lambda_{ik}\right)\left(\prod_{k=1}^{K} d\Lambda_{kk}\Lambda_{p+1,k}\right).$$

The proof consists of three parts: the base case, adding an off-diagonal element and adding an on-diagonal element. For the base-case, note that the expression is correct for $p = 1$ and $K = 0$ (Eq. 19). Next, we add an off-diagonal element, $G_{p+1,K+1}$, where $K+1 < p+1$. We begin by computing $dG_{p+1,K+1}$. Note that the sum only goes to $K+1$, because $\Lambda_{K+1,j} = 0$ for $j > (K+1)$:

$$G_{p+1,K+1} = \sum_{j=1}^{K+1} \Lambda_{p+1,j}\Lambda_{K+1,j},$$
$$dG_{p+1,K+1} = \sum_{j=1}^{K+1} \left(\Lambda_{K+1,j}d\Lambda_{p+1,j} + \Lambda_{p+1,j}d\Lambda_{K+1,j}\right).$$

Remembering that $d\Lambda_{ij}^2 = 0$, the only term that does not cancel when we multiply by the volume element for the previous terms is that for $d\Lambda_{p+1,K+1}$:

$$\underbrace{\left(\prod_{i=1}^{p}\prod_{k=1}^{i} dG_{ik}\right)}_{\text{vol. elem. for } \mathbf{G}_{:p,:p}} \underbrace{\left(\prod_{k=1}^{K+1} dG_{p+1,k}\right)}_{\text{vol. elem. for } \mathbf{G}_{p+1,:K+1}} = \left(\prod_{i=1}^{p}\prod_{k=1}^{i} dG_{ik}\right)\left(\prod_{k=1}^{K} dG_{p+1,k}\right)dG_{p+1,K+1}$$

$$= 2^p \left(\prod_{i=1}^{p}\prod_{k=1}^{i} \Lambda_{kk}d\Lambda_{ik}\right)\left(\prod_{k=1}^{K} d\Lambda_{kk}\Lambda_{p+1,k}\right)dG_{p+1,K+1}$$

$$= 2^p \left(\prod_{i=1}^{p}\prod_{k=1}^{i} \Lambda_{kk}d\Lambda_{ik}\right)\left(\prod_{k=1}^{K} d\Lambda_{kk}\Lambda_{p+1,k}\right)(\Lambda_{K+1,K+1}d\Lambda_{p+1,K+1})$$

$$= 2^p \left(\prod_{i=1}^{p}\prod_{k=1}^{i} \Lambda_{kk}d\Lambda_{ik}\right)\left(\prod_{k=1}^{K+1} d\Lambda_{kk}\Lambda_{p+1,k}\right).$$

So the expression is consistent when adding an on-diagonal element. Finally, the volume element for $G_{p+1,p+1}$ is,

$$G_{p+1,p+1} = \sum_{j=1}^{p+1} \Lambda_{p+1,j}^2 dG_{p+1,K+1} \qquad\qquad = \sum_{j=1}^{K+1} \Lambda_{p+1,j}d\Lambda_{p+1,j}.$$

Remembering that $d\Lambda_{ij}^2 = 0$, the only term that does not cancel when we multiply by the volume element for the previous terms is that for $d\Lambda_{p+1,p+1}$,

$$
\underbrace{\left(\prod_{i=1}^{p+1}\prod_{k=1}^{i} dG_{ik}\right)}_{\text{vol. elem. for } \mathbf{G}_{:p+1,:p+1}} = \underbrace{\left(\prod_{i=1}^{p}\prod_{k=1}^{i} dG_{ik}\right)}_{\text{vol. elem. for } \mathbf{G}_{:p,:p}} \underbrace{\left(\prod_{k=1}^{p+1} dG_{p+1,k}\right)}_{\text{vol. elem. for } \mathbf{G}_{p+1,:p+1}}
$$

$$
= \left(\prod_{i=1}^{p}\prod_{k=1}^{i} dG_{ik}\right) \left(\prod_{k=1}^{p} dG_{p+1,k}\right) dG_{p+1,p+1}
$$

$$
= 2^p \left(\prod_{i=1}^{p}\prod_{k=1}^{i} \Lambda_{kk} d\Lambda_{ik}\right) \left(\prod_{k=1}^{p} d\Lambda_{kk}\Lambda_{p+1,k}\right) dG_{p+1,p+1}
$$

$$
= 2^p \left(\prod_{i=1}^{p}\prod_{k=1}^{i} \Lambda_{kk} d\Lambda_{ik}\right) \left(\prod_{k=1}^{p} d\Lambda_{kk}\Lambda_{p+1,k}\right) (2\Lambda_{p+1,p+1} d\Lambda_{p+1,p+1})
$$

$$
= 2^{p+1} \left(\prod_{i=1}^{p}\prod_{k=1}^{i} \Lambda_{kk} d\Lambda_{ik}\right) \left(\prod_{k=1}^{p+1} d\Lambda_{kk}\Lambda_{p+1,k}\right)
$$

$$
= 2^{p+1} \left(\prod_{i=1}^{p+1}\prod_{k=1}^{i} \Lambda_{kk} d\Lambda_{ik}\right).
$$

Thus, the final result is:

$$
\left(\prod_{i=1}^{P}\prod_{k=1}^{i} dG_{ik}\right) = \left(2^P \prod_{i=1}^{\min(P,\nu)} \Lambda_{ii}^{P+1-i}\right) \left(\prod_{i=1}^{P}\prod_{k=1}^{i} d\Lambda_{ik}\right). \tag{20}
$$

$$
d\mathbf{G} = d\mathbf{\Lambda} \prod_{i=1}^{P} 2\Lambda_{ii}^{P+1-i}. \tag{21}
$$

## B.1 Singular matrices

The above derivation can be extended to the singular case, where $\mathbf{\Lambda}$ has a form mirroring that of $\mathbf{A}$ in Eq. (7a):

$$
\mathbf{\Lambda} = \begin{pmatrix} \Lambda_{11} & \dots & 0 \\ \vdots & \ddots & \vdots \\ \Lambda_{\nu 1} & \dots & \Lambda_{\nu\nu} \\ \vdots & \vdots & \vdots \\ \Lambda_{P1} & \dots & \Lambda_{P\nu} \end{pmatrix}. \tag{22}
$$

To form a valid Jacobian, we need the same number of inputs as outputs. We therefore consider differences in only the corresponding part of $\mathbf{G}$ (i.e. $G_{i,j\leq\min(i,\nu)}$). The recursive expression is

$$
\underbrace{\left(\prod_{i=1}^{p}\prod_{k=1}^{\min(i,\nu)} dG_{ik}\right)}_{\text{vol. elem. for } \mathbf{G}_{:p,:p}} \underbrace{\left(\prod_{k=1}^{K} dG_{p+1,k}\right)}_{\text{vol. elem. for } \mathbf{G}_{p+1,:K}} = 2^{\min(p,\nu)} \left(\prod_{i=1}^{p}\prod_{k=1}^{\min(i,\nu)} \Lambda_{kk} d\Lambda_{ik}\right) \left(\prod_{k=1}^{K} d\Lambda_{kk}\Lambda_{p+1,k}\right),
$$

where $K < \min(p, \nu)$. For $P \leq \nu$, the recursion is exactly as in the full-rank case. For $P > \nu$, the key difference is that there are no longer any on-diagonal elements. As such, for $K = \nu$ we have

$$\left( \prod_{i=1}^{p+1} \prod_{k=1}^{\min(i,\nu)} dG_{ik} \right) = \underbrace{\left( \prod_{i=1}^{p} \prod_{k=1}^{\min(i,\nu)} dG_{ik} \right)}_{\text{vol. elem. for } \mathbf{G}_{:p,:p}} \underbrace{\left( \prod_{k=1}^{\nu} dG_{p+1,k} \right)}_{\text{vol. elem. for } \mathbf{G}_{p+1,:\nu}}$$

$$= 2^{\min(p,\nu)} \left( \prod_{i=1}^{p} \prod_{k=1}^{\min(i,\nu)} \Lambda_{kk} d\Lambda_{ik} \right) \left( \prod_{k=1}^{K} d\Lambda_{kk} \Lambda_{p+1,k} \right)$$

$$= 2^{\min(p,\nu)} \left( \prod_{i=1}^{p+1} \prod_{k=1}^{\min(i,\nu)} \Lambda_{kk} d\Lambda_{ik} \right).$$

The final expression, allowing for the possibility of singular and non-singular matrices, is thus

$$\left( \prod_{i=1}^{P} \prod_{k=1}^{\min(i,\nu)} dG_{ik} \right) = \left( \prod_{i=1}^{\min(P,\nu)} 2\Lambda_{ii}^{P+1-i} \right) \left( \prod_{i=1}^{P} \prod_{k=1}^{\min(i,\nu)} d\Lambda_{ik} \right)$$

$$d\mathbf{G} = d\mathbf{\Lambda} \prod_{i=1}^{\min(P,\nu)} 2\Lambda_{ii}^{P+1-i}.$$

## C  Jacobian for product of two different lower triangular matrices

In this section, we compute the Jacobian for the transformation from $\mathbf{A}$ to $\mathbf{\Lambda} = \mathbf{LA}$. We begin by noting that $\mathbf{\Lambda}$ (Eq. 22) is a potentially rectangular lower-triangular matrix, with the same structure as $\mathbf{A}$. Writing this out,

$$\begin{pmatrix} \Lambda_{11} & 0 & 0 \\ \Lambda_{21} & \Lambda_{22} & 0 \\ \Lambda_{31} & \Lambda_{32} & \Lambda_{33} \\ \Lambda_{41} & \Lambda_{42} & \Lambda_{43} \\ \Lambda_{51} & \Lambda_{52} & \Lambda_{53} \end{pmatrix} = \begin{pmatrix} L_{11} & 0 & 0 & 0 & 0 \\ L_{21} & L_{22} & 0 & 0 & 0 \\ L_{31} & L_{32} & L_{33} & 0 & 0 \\ L_{41} & L_{42} & L_{43} & L_{44} & 0 \\ L_{51} & L_{52} & L_{53} & L_{54} & L_{55} \end{pmatrix} \begin{pmatrix} A_{11} & 0 & 0 \\ A_{21} & A_{22} & 0 \\ A_{31} & A_{32} & A_{33} \\ A_{41} & A_{42} & A_{43} \\ A_{51} & A_{52} & A_{53} \end{pmatrix}.$$

For the first column,

$$\begin{pmatrix} \Lambda_{11} \\ \Lambda_{21} \\ \Lambda_{31} \\ \Lambda_{41} \\ \Lambda_{51} \end{pmatrix} = \begin{pmatrix} L_{11} & 0 & 0 & 0 & 0 \\ L_{21} & L_{22} & 0 & 0 & 0 \\ L_{31} & L_{32} & L_{33} & 0 & 0 \\ L_{41} & L_{42} & L_{43} & L_{44} & 0 \\ L_{51} & L_{52} & L_{53} & L_{54} & L_{55} \end{pmatrix} \begin{pmatrix} A_{11} \\ A_{21} \\ A_{31} \\ A_{41} \\ A_{51} \end{pmatrix},$$

i.e.

$$\mathbf{\Lambda}_{:,1} = \mathbf{L}\mathbf{A}_{:,1}.$$

For the second column,

$$\begin{pmatrix} 0 \\ \Lambda_{22} \\ \Lambda_{32} \\ \Lambda_{42} \\ \Lambda_{52} \end{pmatrix} = \begin{pmatrix} L_{11} & 0 & 0 & 0 & 0 \\ L_{21} & L_{22} & 0 & 0 & 0 \\ L_{31} & L_{32} & L_{33} & 0 & 0 \\ L_{41} & L_{42} & L_{43} & L_{44} & 0 \\ L_{51} & L_{52} & L_{53} & L_{54} & L_{55} \end{pmatrix} \begin{pmatrix} 0 \\ A_{22} \\ A_{32} \\ A_{42} \\ A_{52} \end{pmatrix}.$$

We can eliminate the first row and column of $\mathbf{C}$,

$$\begin{pmatrix} \Lambda_{22} \\ \Lambda_{32} \\ \Lambda_{42} \\ \Lambda_{52} \end{pmatrix} = \begin{pmatrix} L_{22} & 0 & 0 & 0 \\ L_{32} & L_{33} & 0 & 0 \\ L_{42} & L_{43} & L_{44} & 0 \\ L_{52} & L_{53} & L_{54} & L_{55} \end{pmatrix} \begin{pmatrix} A_{22} \\ A_{32} \\ A_{42} \\ A_{52} \end{pmatrix},$$

i.e.

$$\boldsymbol{\Lambda}_{2:,2} = \mathbf{L}_{2:,2:}\mathbf{A}_{2:,2}.$$

For the third column,

$$
\begin{pmatrix} 0 \\ 0 \\ \Lambda_{33} \\ \Lambda_{43} \\ \Lambda_{53} \end{pmatrix} =
\begin{pmatrix}
L_{11} & 0 & 0 & 0 & 0 \\
L_{21} & L_{22} & 0 & 0 & 0 \\
L_{31} & L_{32} & L_{33} & 0 & 0 \\
L_{41} & L_{42} & L_{43} & L_{44} & 0 \\
L_{51} & L_{52} & L_{53} & L_{54} & L_{55}
\end{pmatrix}
\begin{pmatrix} 0 \\ 0 \\ A_{33} \\ A_{43} \\ A_{53} \end{pmatrix},
$$

so we can eliminate the first two rows and columns of $\mathbf{C}$:

$$
\begin{pmatrix} \Lambda_{33} \\ \Lambda_{43} \\ \Lambda_{53} \end{pmatrix} =
\begin{pmatrix}
L_{33} & 0 & 0 \\
L_{43} & L_{44} & 0 \\
L_{53} & L_{54} & L_{55}
\end{pmatrix}
\begin{pmatrix} A_{33} \\ A_{43} \\ A_{53} \end{pmatrix},
$$

i.e.

$$\boldsymbol{\Lambda}_{3:,3} = \mathbf{L}_{3:,3:}\mathbf{A}_{3:,3}.$$

As such, the full computation $\boldsymbol{\Lambda} = \mathbf{L}\mathbf{A}$ can be written as a matrix-vector multiplication by rearranging the columns of $\boldsymbol{\Lambda}$ and $\mathbf{A}$ into a single vector:

$$
\begin{pmatrix} \boldsymbol{\Lambda}_{1:,1} \\ \boldsymbol{\Lambda}_{2:,2} \\ \vdots \\ \boldsymbol{\Lambda}_{\nu:,\nu} \end{pmatrix} =
\begin{pmatrix}
\mathbf{L}_{1:,1:} & \mathbf{0} & \dots & \mathbf{0} \\
\mathbf{0} & \mathbf{L}_{2:,2:} & \dots & \mathbf{0} \\
\vdots & \vdots & \ddots & \vdots \\
\mathbf{0} & \mathbf{0} & \dots & \mathbf{L}_{\nu:,\nu:}
\end{pmatrix}
\begin{pmatrix} \mathbf{A}_{1:,1} \\ \mathbf{A}_{2:,2} \\ \vdots \\ \mathbf{A}_{\nu:,\nu} \end{pmatrix}.
$$

The Jacobian is given by the determinant of the square matrix, and as the matrix is lower-triangular, the determinant can be written in terms of the diagonal elements of $\mathbf{L}$,

$$
\left( \prod_{i=1}^{P} \prod_{k=1}^{\min(i,\nu)} d\Lambda_{ik} \right) = \left( \prod_{i=1}^{P} L_{ii}^{\min(i,\nu)} \right) \left( \prod_{i=1}^{P} \prod_{k=1}^{\min(i,\nu)} dA_{ik} \right).
$$

## D  Proving the singular Bartlett (above) corresponds to the Wishart

We need to change variables to $A_{jj}$ rather than $A_{jj}^2$ [1]:

$$
\begin{aligned}
\mathrm{P}\left(A_{jj}\right) &= \mathrm{P}\left(A_{jj}^2\right) \left| \frac{\partial A_{jj}^2}{\partial A_{jj}} \right|, \\
&= \mathrm{Gamma}\left(A_{jj}^2; \tfrac{\nu-j+1}{2}, \tfrac{1}{2}\right) 2A_{jj}, \\
&= \frac{\left(A_{jj}^2\right)^{(\nu-j+1)/2-1} e^{-A_{jj}^2/2}}{2^{(\nu-j+1)/2}\Gamma\left(\frac{\nu-j+1}{2}\right)} 2A_{jj}, \\
&= \frac{A_{jj}^{\nu-j} e^{-A_{jj}^2/2}}{2^{(\nu-j-1)/2}\Gamma\left(\frac{\nu-j+1}{2}\right)}.
\end{aligned}
$$

Thus, the probability density for $\mathbf{A}$ under the Bartlett sampling operation is

$$
\mathrm{P}\left(\mathbf{A}\right) = \underbrace{\prod_{j=1}^{\tilde{\nu}} \frac{A_{jj}^{\nu-j} e^{-T_{jj}^2/2}}{2^{\frac{\nu-j-1}{2}}\Gamma\left(\frac{\nu-j+1}{2}\right)}}_{\text{on-diagonals}} \underbrace{\prod_{i=j+1}^{p} \frac{1}{\sqrt{2\pi}} e^{-A_{ij}^2/2}}_{\text{off-diagonals}}, \tag{23}
$$

---

[1] djalil.chafai.net/blog/2015/10/20/bartlett-decomposition-and-other-factorizations/

where $\tilde{\nu} = \min(\nu, P)$. To convert this to a distribution on $\mathbf{G}$, we need the volume element for the transformation from $\mathbf{A}$ to $\mathbf{Z} = \mathbf{A}\mathbf{A}^T$, which is given in Appendix B (Eq. (21)):

$$d\mathbf{Z} = d\mathbf{A} \prod_{j=1}^{\tilde{\nu}} 2A_{jj}^{P-j+1}.$$

Thus

$$P(\mathbf{Z}) = P(\mathbf{A}) \left( \prod_{j=1}^{\tilde{\nu}} \tfrac{1}{2} A_{jj}^{-(P-j+1)} \right)$$

$$= \prod_{j=1}^{\tilde{\nu}} \frac{A_{jj}^{\nu-P-1} e^{-T_{jj}^2/2}}{2^{\frac{\nu-j+1}{2}} \Gamma\left(\frac{\nu-j+1}{2}\right)} \prod_{i=j+1}^{P} \frac{1}{\sqrt{2\pi}} e^{-A_{ij}^2/2}.$$

Now, we break this expression down into separate components and perform standard algebraic manipulations. First, we manipulate a product over the diagonal elements of $\mathbf{A}$ to obtain the determinant of $\mathbf{Z}$:

$$\prod_{j=1}^{\tilde{\nu}} A_{jj}^{\nu-P-1} = \left( \prod_{j=1}^{\tilde{\nu}} A_{jj} \right)^{\nu-P-1} = \left| \mathbf{A}_{:\tilde{\nu},:} \mathbf{A}_{:\tilde{\nu},:}^T \right|^{(\nu-P-1)/2} = \left| \mathbf{Z}_{:\tilde{\nu},:\tilde{\nu}} \right|^{(\nu-P-1)/2}.$$

Next, we manipulate the exponential terms to form an exponentiated trace. We start by combining on- and off-diagonal terms, and noting that $A_{ij} = 0$ for $i < j$, we can extend the sum

$$\prod_{j=1}^{\tilde{\nu}} e^{-A_{jj}^2/2} \prod_{i=j+1}^{P} e^{-A_{ij}^2/2} = \prod_{j=1}^{\tilde{\nu}} \prod_{i=j}^{P} e^{-A_{ij}^2/2} = \prod_{j=1}^{\tilde{\nu}} \prod_{i=1}^{P} e^{-A_{ij}^2/2}.$$

Then we take the product inside the exponential and note that as $\mathbf{Z} = \mathbf{A}\mathbf{A}^T$, we can write the sum as a trace of $\mathbf{Z}$,

$$= e^{\sum_{j=1}^{\tilde{\nu}} \sum_{i=1}^{P} -A_{ij}^2/2} = e^{-\operatorname{Tr}(\mathbf{Z})/2}.$$

Next, we consider powers of 2. We begin by computing the number of $1/\sqrt{2}$ terms, arising from the off-diagonal elements,

$$\prod_{j=1}^{\tilde{\nu}} \prod_{i=j+1}^{p} \frac{1}{\sqrt{2}} = \left( \frac{1}{\sqrt{2}} \right)^{\nu(p-\tilde{\nu})+\tilde{\nu}(\tilde{\nu}-1)/2}.$$

Note that the $\tilde{\nu}(\tilde{\nu}-1)$ term corresponds to the off-diagonal terms in the square block $\mathbf{A}_{:\tilde{\nu},:}$, and the $\nu(p-\tilde{\nu})$ term corresponds to the terms from $\mathbf{A}_{\tilde{\nu}:,:}$. Next we consider the on-diagonal terms,

$$\prod_{j=1}^{\tilde{\nu}} \frac{1}{2^{(\nu-j+1)/2}} = \left( \frac{1}{\sqrt{2}} \right)^{\tilde{\nu}(\nu+1)} \prod_{j=1}^{\tilde{\nu}} \left( \frac{1}{\sqrt{2}} \right)^{-j} = \left( \frac{1}{\sqrt{2}} \right)^{\tilde{\nu}(\nu+1)-\tilde{\nu}(\tilde{\nu}+1)/2}.$$

Combining the on and off-diagonal terms,

$$\prod_{j=1}^{\tilde{\nu}} \frac{1}{2^{(\nu-j+1)/2}} \prod_{i=j+1}^{P} \frac{1}{\sqrt{2}} = \left( \frac{1}{\sqrt{2}} \right)^{\nu(P-\tilde{\nu})+\tilde{\nu}(\tilde{\nu}-1)/2+\tilde{\nu}(\nu+1)-\tilde{\nu}(\tilde{\nu}+1)/2}$$

$$= \left( \frac{1}{\sqrt{2}} \right)^{(\nu P-\nu\tilde{\nu})+(\tilde{\nu}^2/2-\tilde{\nu}/2)+(\tilde{\nu}\nu+\tilde{\nu})+(-\tilde{\nu}^2/2-\tilde{\nu}/2)}$$

$$= \left( \frac{1}{\sqrt{2}} \right)^{\nu P}.$$

Finally, using the definition of the multivariate Gamma function,

$$\prod_{j=1}^{\tilde{\nu}} \Gamma\left(\tfrac{\nu-j+1}{2}\right) \prod_{i=j+1}^{P} \sqrt{\pi} = \pi^{\nu(P-\tilde{\nu})/2} \pi^{\tilde{\nu}(\tilde{\nu}-1)/4} \underbrace{\prod_{j=1}^{\tilde{\nu}} \Gamma\left(\tfrac{\nu-j+1}{2}\right)}_{=\Gamma_{\tilde{\nu}}\left(\frac{\nu}{2}\right)}$$

$$= \pi^{\nu(P-\tilde{\nu})/2} \Gamma_{\tilde{\nu}}\left(\tfrac{\nu}{2}\right).$$

We thereby re-obtain the probability density for the standard Wishart distribution,

$$P\left(\mathbf{Z}\right) = \frac{\pi^{\nu(\tilde{\nu}-P)/2}}{2^{\nu P/2}\Gamma_{\tilde{\nu}}\left(\frac{\nu}{2}\right)} \left|\mathbf{Z}_{:\tilde{\nu},:\tilde{\nu}}\right|^{(\nu-P-1)/2} e^{-\operatorname{Tr}(\mathbf{Z})/2}.$$

For $\tilde{\nu} = \nu$, this matches Eq. 3.2 in Srivastava et al. (2003), and for $\tilde{\nu} = P$ it matches the standard full-rank Wishart probability density function.

## E  Choice of $\mathbf{F_i}$

Here, we establish that the distribution over $\mathbf{F_t F_i}^T$ and $\mathbf{F_t F_t}^T$ does not depend on the choice of $\mathbf{F_i}$. Due to the definition of $\mathbf{F}_t$ (Eq. 17) we can write,

$$\mathbf{F}_t = \mathbf{K}_{\mathrm{ti}}^T \mathbf{K}_{\mathrm{ii}}^{-1}\mathbf{F_i} + \mathbf{K}_{\mathrm{tt \cdot i}}^{1/2}\boldsymbol{\Xi}.$$

where $\boldsymbol{\Xi}$ is a matrix with IID standard Gaussian elements. Thus,

$$\mathbf{F_t F_i}^T = \mathbf{K}_{\mathrm{ti}}^T \mathbf{K}_{\mathrm{ii}}^{-1}\mathbf{F_i F_i}^T + \mathbf{K}_{\mathrm{tt \cdot i}}^{1/2}\boldsymbol{\Xi}\mathbf{F_i}^T$$

$$\mathbf{F_t F_i}^T \sim \mathcal{MN}\left(\mathbf{K}_{\mathrm{ti}}^T \mathbf{K}_{\mathrm{ii}}^{-1}\mathbf{G}_{\mathrm{ii}}, \mathbf{K}_{\mathrm{tt \cdot i}}, \mathbf{G}_{\mathrm{ii}}\right).$$

We can do the same for $\mathbf{F_t F_t}^T$:

$$\begin{aligned}
\mathbf{F_t F_t}^T = {}&\mathbf{K}_{\mathrm{ti}}^T \mathbf{K}_{\mathrm{ii}}^{-1}\mathbf{F_i F_i}^T \mathbf{K}_{\mathrm{ii}}^{-1}\mathbf{K}_{\mathrm{ti}} \\
&+ \mathbf{K}_{\mathrm{ti}}^T \mathbf{K}_{\mathrm{ii}}^{-1}\mathbf{F_i}\boldsymbol{\Xi}\mathbf{K}_{\mathrm{tt \cdot i}}^{1/2} + \mathbf{K}_{\mathrm{tt \cdot i}}^{1/2}\boldsymbol{\Xi}\mathbf{F_i}^T \mathbf{K}_{\mathrm{ii}}^{-1}\mathbf{K}_{\mathrm{ti}} \\
&+ \mathbf{K}_{\mathrm{tt \cdot i}}^{1/2}\boldsymbol{\Xi}\boldsymbol{\Xi}\mathbf{K}_{\mathrm{tt \cdot i}}^{1/2}.
\end{aligned}$$

The first term is independent of the choice of of $\mathbf{F_i}$ because $\mathbf{G}_{\mathrm{ii}} = \mathbf{F_i F_i}^T$. The term on the last line does not depend on $\mathbf{F_i}$ at all. Finally, the two terms in the middle are Gaussian with covariance that depends on $\mathbf{G}_{\mathrm{ii}}$ but not the specific choice of $\mathbf{F_i}$:

$$\mathbf{K}_{\mathrm{ti}}^T \mathbf{K}_{\mathrm{ii}}^{-1}\mathbf{F_i}\boldsymbol{\Xi}\mathbf{K}_{\mathrm{tt \cdot i}}^{1/2} \sim \mathcal{MN}\left(\mathbf{0}, \mathbf{K}_{\mathrm{ti}}^T \mathbf{K}_{\mathrm{ii}}^{-1}\mathbf{G_i}\mathbf{K}_{\mathrm{ii}}^{-1}\mathbf{K}_{\mathrm{ti}}, \mathbf{K}_{\mathrm{tt \cdot i}}\right),$$

$$\mathbf{K}_{\mathrm{tt \cdot i}}^{1/2}\boldsymbol{\Xi}\mathbf{F_i}^T \mathbf{K}_{\mathrm{ii}}^{-1}\mathbf{K}_{\mathrm{ti}} \sim \mathcal{MN}\left(\mathbf{0}, \mathbf{K}_{\mathrm{tt \cdot i}}, \mathbf{K}_{\mathrm{ti}}^T \mathbf{K}_{\mathrm{ii}}^{-1}\mathbf{G_i}\mathbf{K}_{\mathrm{ii}}^{-1}\mathbf{K}_{\mathrm{ti}}\right).$$

Thus, the additional components of $\mathbf{G}$, $\mathbf{G}_{\mathrm{ti}} = \mathbf{F_t F_i}$ and $\mathbf{G}_{\mathrm{tt}} = \mathbf{F_t F_t}$ depend on $\mathbf{G}_{\mathrm{ii}}$ but not on the specific choice of $\mathbf{F_i}$. Thus, any $\mathbf{F_i}$ can be used as long as $\mathbf{G}_{\mathrm{ii}} = \mathbf{F_i F_i}^T$.

## F  Experimental details

**Datasets**    All experiments were performed using the UCI splits from Gal & Ghahramani (2015), available at `https://github.com/yaringal/DropoutUncertaintyExps/tree/master/UCI_Datasets`. For each dataset there are twenty splits, with the exception of protein, which only has five. We report mean plus or minus one standard error over the splits.

**Model details**    As standard, we set $\nu$ (the 'width' of each layer) to be equal to the dimensionality of the input space. We use the squared exponential kernel, with automatic relevance determination (ARD) in the first layer, but without for the intermediate layers as ARD relies on explicit features existing. However, we found in practice that using ARD for intermediate layers in a DGP did not hugely affect the results, as each output GP in a layer shares the same prior and hence output prior variance. For the final, GP, layer of the DWP model we use a global inducing approximate posterior (Ober & Aitchison, 2021), as done in the DGP. We leave the particular implementation details for the code provided with the paper, but we note that we use the 'sticking the landing' gradient estimator (Roeder et al., 2017) for the $\{\boldsymbol{\alpha}_\ell, \boldsymbol{\beta}_\ell, \boldsymbol{\mu}_\ell, \boldsymbol{\sigma}_\ell\}_{\ell=1}^L$ approximate posterior parameters of the DWP (using it for the other parameters, as well as for the DGP parameters, is difficult as the parameters of one layer will affect the KL estimate of the following layers for global inducing posteriors).

**Training details**    We train all models using the same training scheme. We use 20,000 gradient steps to train each model, using the Adam optimizer (Kingma & Ba, 2015) with an initial learning rate of 1e-2. We anneal the KL using a factor increasing linearly from 0 to 1 over the first 1,000 gradient steps, and step the learning rate down to 1e-3 after 10,000 gradient steps. We use 10 samples from the approximate posterior for training, and 100 for testing. Experiments were performed using an internal cluster of machines with NVIDIA GeForce 2080 Ti GPUs, although we used CPU (Intel Core i9-10900X) for the smaller datasets (boston, concrete, energy, wine, yacht).

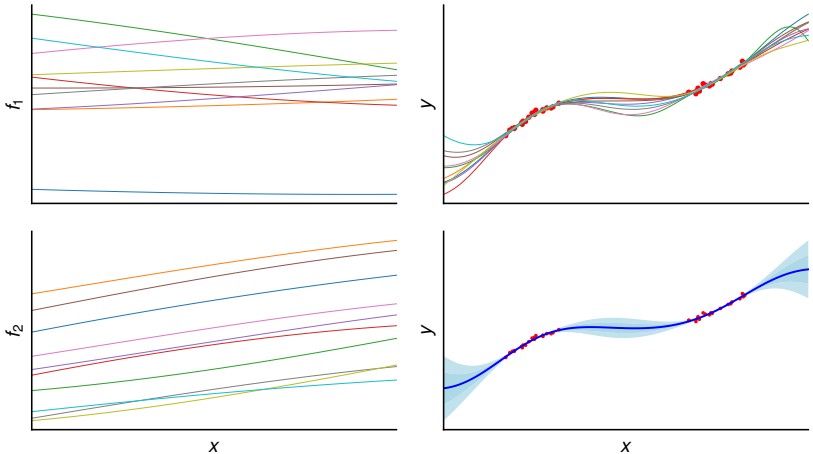

Figure 2: Features from a 2-layer DGP posterior with intermediate width 2: feature samples $f_1$ (first layer, first output; top left), $f_2$ (first layer, second output; bottom left), posterior samples (top right), posterior predictive (bottom right)

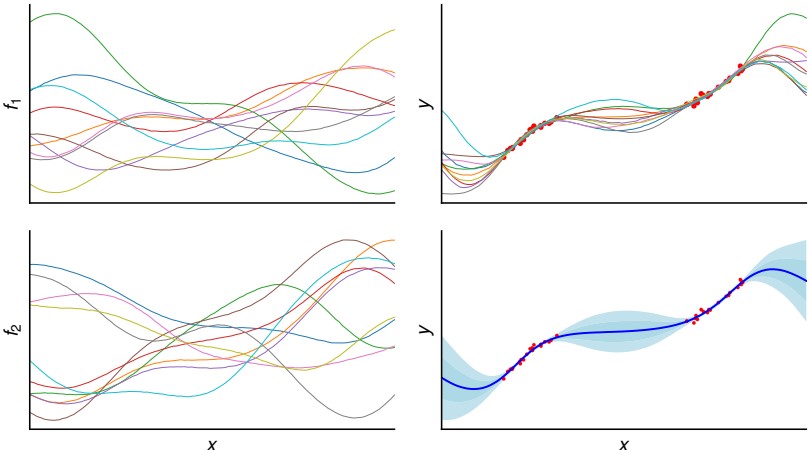

Figure 3: Features from a 2-layer DWP posterior with intermediate width 2: feature samples $f_1$ (first layer, first output; top left), $f_2$ (first layer, second output; bottom left), posterior samples (top right), posterior predictive (bottom right)

## G   Toy comparison of DGP and DWP posteriors

In this section, we compare intermediate-layer features for trained 2-layer, width-2 DWP and DGP models on a 1-dimensional toy example (following the toy example from Ober & Aitchison (2021)). We plot the intermediate samples for the DGP in Figure 2 and those for the DWP in Figure 3. We observe that the features learned by the DWP are both more interesting, and more varied. This allows for a greater predictive uncertainty in the posterior away from the data. These improved characteristics are due to the increased ability of the DWP to capture true-posterior symmetries.

# H  Tables

In Table 3, we include a comparison to the reported results for the 3-layer DIWP with squared exponential kernel from Aitchison et al. (2021); they did not provide ELBOs. However, it should be noted that the specific implementation and architectural details differ significantly from those presented in this paper, and so these results are not directly comparable. Additionally, Aitchison et al. (2021) bases its error bars on paired comparisons to the other methods instead of the standard error bars we use here; we therefore omit the error bars completely.

Table 2: ELBOs per datapoint. We report mean plus or minus one standard error over the splits.

| {dataset} - {depth} | DWP | DGP |
|---|---|---|
| boston - 2 | **-0.33 $\pm$ 0.00** | -0.38 $\pm$ 0.01 |
| 3 | **-0.34 $\pm$ 0.01** | -0.40 $\pm$ 0.01 |
| 4 | **-0.36 $\pm$ 0.01** | -0.43 $\pm$ 0.00 |
| 5 | **-0.38 $\pm$ 0.01** | -0.45 $\pm$ 0.01 |
| concrete - 2 | **-0.42 $\pm$ 0.00** | -0.44 $\pm$ 0.00 |
| 3 | **-0.43 $\pm$ 0.00** | -0.47 $\pm$ 0.00 |
| 4 | **-0.46 $\pm$ 0.00** | -0.50 $\pm$ 0.00 |
| 5 | **-0.49 $\pm$ 0.00** | -0.50 $\pm$ 0.00 |
| energy - 2 | **1.46 $\pm$ 0.00** | 1.44 $\pm$ 0.00 |
| 3 | **1.44 $\pm$ 0.00** | 1.42 $\pm$ 0.00 |
| 4 | **1.42 $\pm$ 0.00** | 1.40 $\pm$ 0.00 |
| 5 | **1.41 $\pm$ 0.00** | 1.39 $\pm$ 0.00 |
| kin8nm - 2 | -0.16 $\pm$ 0.00 | **-0.15 $\pm$ 0.00** |
| 3 | -0.15 $\pm$ 0.00 | **-0.14 $\pm$ 0.00** |
| 4 | -0.14 $\pm$ 0.00 | -0.14 $\pm$ 0.00 |
| 5 | -0.14 $\pm$ 0.00 | **-0.13 $\pm$ 0.00** |
| naval - 2 | 3.75 $\pm$ 0.07 | **3.91 $\pm$ 0.08** |
| 3 | 3.76 $\pm$ 0.11 | 3.82 $\pm$ 0.10 |
| 4 | 3.68 $\pm$ 0.07 | **3.91 $\pm$ 0.06** |
| 5 | 3.65 $\pm$ 0.07 | **3.91 $\pm$ 0.10** |
| power - 2 | 0.03 $\pm$ 0.00 | 0.03 $\pm$ 0.00 |
| 3 | 0.03 $\pm$ 0.00 | 0.03 $\pm$ 0.00 |
| 4 | 0.03 $\pm$ 0.00 | 0.03 $\pm$ 0.00 |
| 5 | **0.03 $\pm$ 0.00** | 0.02 $\pm$ 0.00 |
| protein - 2 | -1.07 $\pm$ 0.00 | -1.07 $\pm$ 0.00 |
| 3 | -1.04 $\pm$ 0.00 | -1.04 $\pm$ 0.00 |
| 4 | -1.02 $\pm$ 0.00 | -1.02 $\pm$ 0.00 |
| 5 | -1.01 $\pm$ 0.00 | **-1.00 $\pm$ 0.00** |
| wine - 2 | -1.18 $\pm$ 0.00 | -1.18 $\pm$ 0.00 |
| 3 | **-1.18 $\pm$ 0.00** | -1.19 $\pm$ 0.00 |
| 4 | **-1.18 $\pm$ 0.00** | -1.19 $\pm$ 0.00 |
| 5 | -1.19 $\pm$ 0.00 | -1.19 $\pm$ 0.00 |
| yacht - 2 | **2.02 $\pm$ 0.01** | 1.80 $\pm$ 0.03 |
| 3 | **1.89 $\pm$ 0.02** | 1.59 $\pm$ 0.01 |
| 4 | **1.74 $\pm$ 0.02** | 1.48 $\pm$ 0.02 |
| 5 | **1.65 $\pm$ 0.01** | 1.37 $\pm$ 0.03 |

Table 3: Average test log likelihoods. We report mean plus or minus one standard error over the splits, along with quoted results for the DIWP model from Aitchison et al. (2021). We only directly compare between DWP and DGP models and do not quote error bars for the DIWP due to the differences noted above.

| {dataset} - {depth} | DWP | DGP | |
|---|---|---|---|
| boston - 2 | $-2.39 \pm 0.05$ | $-2.42 \pm 0.05$ | - |
| 3 | $-2.38 \pm 0.04$ | $-2.41 \pm 0.05$ | - |
| 4 | $-2.38 \pm 0.04$ | $-2.41 \pm 0.04$ | -2.40 |
| 5 | $-2.39 \pm 0.04$ | $-2.44 \pm 0.04$ | - |
| concrete - 2 | $-3.13 \pm 0.02$ | $-3.10 \pm 0.02$ | - |
| 3 | $-3.11 \pm 0.02$ | $-3.08 \pm 0.02$ | - |
| 4 | $-3.12 \pm 0.02$ | $-3.11 \pm 0.02$ | -3.08 |
| 5 | $-3.13 \pm 0.01$ | $-3.14 \pm 0.02$ | - |
| energy - 2 | $-0.70 \pm 0.03$ | $-0.70 \pm 0.03$ | - |
| 3 | $-0.71 \pm 0.03$ | $-0.70 \pm 0.03$ | - |
| 4 | $-0.70 \pm 0.03$ | $-0.70 \pm 0.03$ | - 0.70 |
| 5 | $-0.70 \pm 0.03$ | $-0.70 \pm 0.03$ | - |
| kin8nm - 2 | $1.35 \pm 0.00$ | $1.35 \pm 0.00$ | - |
| 3 | $1.37 \pm 0.00$ | $1.38 \pm 0.01$ | - |
| 4 | $\mathbf{1.40 \pm 0.00}$ | $1.38 \pm 0.01$ | 1.01 |
| 5 | $1.40 \pm 0.01$ | $1.39 \pm 0.01$ | - |
| naval - 2 | $8.11 \pm 0.10$ | $8.24 \pm 0.08$ | - |
| 3 | $8.22 \pm 0.07$ | $8.13 \pm 0.14$ | - |
| 4 | $8.18 \pm 0.05$ | $8.27 \pm 0.05$ | 5.92 |
| 5 | $8.21 \pm 0.05$ | $8.31 \pm 0.07$ | - |
| power - 2 | $-2.77 \pm 0.01$ | $-2.78 \pm 0.01$ | - |
| 3 | $-2.77 \pm 0.01$ | $-2.77 \pm 0.01$ | - |
| 4 | $-2.77 \pm 0.01$ | $-2.78 \pm 0.01$ | -2.78 |
| 5 | $-2.77 \pm 0.01$ | $-2.78 \pm 0.01$ | - |
| protein - 2 | $\mathbf{-2.81 \pm 0.00}$ | $-2.82 \pm 0.00$ | - |
| 3 | $-2.78 \pm 0.00$ | $\mathbf{-2.77 \pm 0.00}$ | - |
| 4 | $-2.73 \pm 0.01$ | $-2.75 \pm 0.01$ | -2.74 |
| 5 | $-2.73 \pm 0.00$ | $-2.73 \pm 0.00$ | - |
| wine - 2 | $-0.96 \pm 0.01$ | $-0.96 \pm 0.01$ | - |
| 3 | $-0.96 \pm 0.01$ | $-0.96 \pm 0.01$ | - |
| 4 | $-0.96 \pm 0.01$ | $-0.96 \pm 0.01$ | -1.00 |
| 5 | $-0.96 \pm 0.01$ | $-0.96 \pm 0.01$ | - |
| yacht - 2 | $\mathbf{-0.04 \pm 0.09}$ | $-0.43 \pm 0.10$ | - |
| 3 | $\mathbf{-0.16 \pm 0.07}$ | $-0.65 \pm 0.04$ | - |
| 4 | $\mathbf{-0.43 \pm 0.10}$ | $-0.70 \pm 0.04$ | -0.39 |
| 5 | $\mathbf{-0.43 \pm 0.10}$ | $-0.88 \pm 0.08$ | - |

Table 4: Average runtime (seconds) for an epoch of boston and protein. Error bars are negligible.

| {dataset} - {depth} | DWP | DGP |
|---|---|---|
| boston - 2 | 0.214 | 0.318 |
| 5 | 0.351 | 0.668 |
| protein - 2 | 0.942 | 0.987 |
| 5 | 1.815 | 2.017 |