# OpenReview forum: "A variational approximate posterior for the deep Wishart process"
_NeurIPS.cc/2021/Conference — NeurIPS 2021 Poster_

### Official Review · Reviewer_uQwZ · 2021-07-14

**Rating:** 6
**Confidence:** 4

**Summary:**

Deep kernel process (DKP, Aitchison et al., 2020) is a new probabilistic model that provides more flexible modelling power than a shallow kernel method. The deep Wishart process is equivalent to deep Gaussian process, but there is no inference algorithm in the original paper. This paper proposes a new variational inference algorithm for it. It proposes a generalised singular Wishart distribution to model the posterior distribution of the random Gram matrix in DWPs and apply the doubly-stochstic inducing-point scheme. This is a nice contribution to the progress of research along the line of DKPs.

**Limitations And Societal Impact:**

The authors answers "Yes" to question 1.b "Did you describe the limitations of your work?" but I don't see where the limitations are discussed.

**Main Review:**

Significance & originality:

DKPs have a good property of preserving the symmetries in latent features that standard variational inference for DGPs does not. Earlier work developed variational inference method for deep inverse Wishart processes which has a different prior distribution from deep GPs. This work provides a variational inference method for deep Wishard processes so that one can compare the performance of DWPs with DGPs side by side. This is a nice contribution to the research in DKPs. The main novelty of this inference method is introducing the generalised Wishart distribution, while the application of doubly-stochstic inducing-point inference is mostly standard.

To better assess the significance of introducing the inference algorithm for DWPs, I think it would be useful to compare also the result of DIWPs. After all, we would like to find a flexible probabilistic model to have a high log-likelihood (or approximately ELBO) on train and test set.


Quality:

The derivation of the generalised Wishart distribution based on Bartlett decomposition and its application in the variational inference algorithm looks good to me although I haven't checked the detailed derivation in the appendix.

I have a question about the use of generalised Wishart distribution as the approximate posterior distribution. Do we know if the true posterior distribution of the Gram matrix has at most \nu ranks or it could actually be higher than that? If the rank in the posterior could go higher, shall we consider the \nu parameter in Q as another hyper-parameter to tune? Besides, could the authors comment on the capacity of the generalised Wishart distribution? How does the flexibility / capacity compare with the Gaussian approximation in DGPs besides its ability to model symmetries?


Clarity:

The authors provide sufficient background to understand deep Wishart processes and the proposed approximate posterior distribution. The methods section also provides an appropriate level of details in the main text to understand its algorithm.

For the experiment section, I would appreciate more analysis of the inference results beyond simply comparing the ELBO and test log-likelihood. It would be very helpful if the authors could show the difference in the inferred posterior using DWP and DGP. After all, the advantage of using DWPs versus DIWPs is that we can have a more direct comparison with DGPs but Table 1 doesn't give much information beside a better ELBO score.

The authors mentioned the computational complexity is lower than the for DGPs because it doesn't have to sample for each latent feature dimension separately. Does it mean the runtime of the proposed method is indeed lower than that for DGPs? If so, it would also be nice to show the improvement of the runtime in the experiments.

**Time Spent Reviewing:**

4

---

> ### Author Response · Authors · 2021-08-10
> **Response**
>
> Thank you for your positive, thoughtful and insightful comments! In response to your points:
>
> **DIWP Comparison**: We have added a comparison between the ELBO and test log likelihoods reported here with those in the original DKP paper for the DIWP.  We can see that DWP "wins" on kin8nm, naval, wine and protein while DIWP "wins" on concrete, where we have attempted to assess significance using the error bars from this submission.  That said, we believe the real value of this work is as a stepping-stone on the path towards convolutional DKPs.  To achieve that, we need efficient low-rank approximations, which we believe are only possible in the low-rank DWP rather than the full-rank DIWP, hence our development of the DWP in this paper (which would be too much for one paper).
>
> **Posterior rank**: Good question! The true posterior over the Gram matrix is at most rank nu, as the prior assigns zero probability to matrices with higher rank. The rank of all the Gram matrices is therefore fixed by our choice of nu in the prior.  Sadly, nu can't be trained using gradient descent as it is a non-negative integer, so it does need to be treated as an extra hyperparameter, analogous to DGP/NN width.  In our case, we choose it to be the same as the conventional width of a DGP from Salimbeni and Diesenroth (2017).  We have clarified this in the main text.
>
> **Flexibility/capacity**: In general, assessing flexibility/capacity is difficult, as we don't know of any formal definitions.  We can say that, in principle, the flexibility/capacity of DWP priors should be equivalent to that of the DGP prior, as they induce the same distribution over functions.  In practice, the flexibility/capacity can be limited by a poor choice of approximate posterior.  Our good practical performance indicates that we probably aren't being limited by a poor choice of approximate posterior here.  But we expect that there is still room to improve the approximate posterior.
>
> **Posterior**: To compare the posteriors in more detail, we have added plots of the posterior over intermediate layer features implied by the DWP, compared to the equivalent DGP.  As expected, there is far more flexibility in the intermediate layer features for the DWP, as the DWP respects unitary symmetries in the true posterior features (Eq. 16 and 17).
>
> **Runtime**: Great suggestion!  In our experiments, the DWP has lower runtime than DGPs (see also reply to R1): for example 5-layer DWPs take 0.368 seconds per epoch for boston, whereas the equivalent DGP takes 0.713 seconds per epoch.  As we use 20,000 gradient steps for each training run for both DGPs and DWPs, there are similar improvements in the overall length of the experiment.  We will include a detailed breakdown of the runtimes for different datasets, and plots of the performance vs gradient steps and vs wallclock time.
>
> **Limitations**: We have added a section on the limitations, with the key limitation being that the generalisation to neural network architectures such as convolutional or graph neural networks is non-trivial, and requires considerable additional research.

---

### Official Review · Reviewer_qTiX · 2021-07-15

**Rating:** 5
**Confidence:** 3

**Summary:**

The manuscript proposes a distribution over positive semi-definite matrices based on the singular Bartlett decomposition for a deep Wishart process (DWP), a special case of deep kernel learning [Aitchison et al., 2020] where gram matrices are used as the latent variables and are transformed through nonlinear kernel functions layer by layer. Based on this distribution, the authors developed a doubly stochastic inducing inference method for the DWP (previously inference in a DWP was not available) by adapting the corresponding variational inference method in [Aitchison et al., 2020].



**Limitations And Societal Impact:**

Have the authors adequately addressed the limitations and potential negative societal impact of their work? Yes.

**Main Review:**


(Originality & quality) I think that the manuscript has certain contributions, one of which is developing the variational inference method for the DWP by proposing a distribution over positive semi-definite matrices based on the singular Bartlett decomposition. Although the propose method is not super new, the method does not seem to be trivial. I did not check every derivation but the method looks technically sound.

(Significance)  However, I am concerned that this work might not attract broad interest. The manuscript proposes an inference method (which is not entirely different or new from previous work) to a specific model formulation. In addition, the experimental results section includes only toy datasets and a DGP as a comparison method. It is understandable that the manuscript attempts comparing the proposed method to the corresponding DGP model based on the connection between both formulations, but at the same time this approach would limit the significance of the proposed work. Readers cannot be convinced about the effectiveness of the proposed method to real-world problems.

Minor comments
It was at the first read a little bit difficult to understand how the proposed distribution (eq. 27) works since the scale parameter \bSigma does not appear in the distribution. It would be helpful if there was a reminder that the Cholesky \bL is calculated from the scale parameter around eq. 27.

**Time Spent Reviewing:**

6 hours

---

> ### Author Response · Authors · 2021-08-10
> **Response**
>
> Thank you for your review! We first address your comments on the originality of our work, followed by discussing its significance.
>
> Originality
> -----
> Our paper describes the first inference scheme for deep Wishart processes.  This is highly non-trivial: the authors of the original DKP paper described how they would have preferred to work with DWPs because of the exact prior equivalence to DGPs. However, those authors go on to explicitly describe how they were unable develop a suitable approximate posterior for DWPs and so were forced to work with DIWPs.
>
> Of course, like all other academic work, we build on ideas from the literature.  But, to build this inference scheme, we needed to make two highly non-trivial technical contributions:
> 1. A distribution over positive semi-definite matrices that allows independent control over the mean and variance, which has a tractable probability density function.  (The original DKP describes how the natural choice is the non-central Wishart, but the pdf turned out to be too complex to implement).  If we were able to find such a distribution, we would have just used it!  Of course, to obtain this distribution, we used methods from the literature (Jacobians and such), but the actual result is to our knowledge entirely novel.
> 2. Sampling test/train points conditioned on the inducing points is entirely different for Wishart and inverse Wishart processes, and thus had to be developed from scratch.
>
> Significance
> -----
> DWPs as developed in our submission have value right now as a replacement for DGPs and small, fully connected NNs.  In addition, better approximate posteriors possible in DKPs will lead reasonably directly to better generalisation bounds via PAC-Bayes. That said, we agree that while DKPs are an exciting emerging area they do require further development before they become very widely used.  We see this paper as a stepping-stone on that path.  In particular, ultimately we want convolutional DKPs for e.g. image classification.  To achieve that, we need efficient low-rank approximations, which we believe are only possible in the low-rank DWP rather than the full-rank DIWP, hence our development of the DWP in this paper (which would be too much for one conference paper).
>
> Finally, on the experimental side, we note that the UCI datasets are standard datasets that DGP methods are usually evaluated on, and consist of either real-world datasets or simulated data for a real-world application. That being said, if you have particular experiments in mind that you would like to see, we would happily consider them!
>
> Minor Comments
> ----
> We have clarified the dependency on \bS in Eq. 27.

---

### Official Review · Reviewer_UZcb · 2021-07-16

**Rating:** 5
**Confidence:** 3

**Summary:**

The authors introduce an approximate inference procedure for the deep Wishart process. In order to do so, they define a generalised singular Wishart process and use this as a variational posterior. The DWP appears to be a model consisting of layers of Wishart process, which are distributions over PSD matrices. This allows one to define a Wishart process whose parameters are the output of another Wishart process. When the a deep GP is constructed with a kernel that is isotropic, the, DGP is a DWP.  However, the usual approximate posterior used in the DGP fails to capture a certain orthogonal invariance property in the features that is present in the true posterior. The DWP on the other hand, deals with Gram matrices instead of features and this requirement from the approximate posterior is automatically satisfied.

**Limitations And Societal Impact:**

N/A.

**Main Review:**

The motivation for why the DWP is superior to the DGP seems to be that the variational posterior that it has access to can encode the symmetry constraint in the posterior. Excluding this initial motivation and the introduced methods, there is very little theory giving any kind of theoretical guarantees. I am also not convinced that this motivation is sufficient to justify an entirely new inference scheme, given that the current empirical evaluation is a bit limited. Are there any other reasons why we would prefer to do inference with a DWP versus a DGP?


Questions/Suggestions for improvement:
1. As currently written, the paper seems to assume a certain familiarity with the original deep kernel processes in its motivation, which is not (yet) a mainstream model. Since the paper is specifically about the deep Wishart process, apparently a special case of the deep kernel process, in order to make the message more direct to the reader I would suggest only briefly mentioning deep kernel processes and then focussing on the DWP.
2. The paper is not the easiest to read. I enjoyed section 2. In section 3, I would prefer a self-contained theorem characterising the generalised singular Wishart distribution, which for the reader doing their first pass, could be used as a abstract tool to understand the rest of the paper. Do all the equations need to be numbered? It seems like many are never referred to in the main text.
3. Experiments. You only compared the DWP to a DGP with squared exponential kernel. Arguably you have removed the flexibility that the DGP has in that it can use non-isotropic kernels. For a fair comparison, the DGP should have access to non-isotropic kernels.
4. I believe the authors should be more upfront about the requirement that the kernel is isotropic in the abstract, as this is a relatively strong constraint. The authors state that "We mainly consider isotropic kernels", but as far as I can tell, they exclusively consider isotropic kernels.


**Time Spent Reviewing:**

2.5

---

> ### Author Response · Authors · 2021-08-10
> **Response**
>
> Thank you for your comments! We first address the question that you gave regarding the motivation before discussing your numbered points.
>
> Capturing the symmetry constraint means we do a _much_ better job of capturing the true posterior, and this is captured in the considerable empirical improvements in the ELBOs.  Interestingly, most PAC-Bayes bounds - just like the ELBO - trade off a predictive performance term against a KL term (e.g. see Germain et al. 2016 "PAC-Bayesian Theory Meets Bayesian Inference").  As such, if we wanted guarantees, our improved ELBOs/approximate posteriors could be used to give better PAC-Bayes generalisation bounds (though this is out-of-scope for the present work).
>
>
> 1. Agreed: the background section already focuses entirely on DWPs, and we have refocused the introduction onto DWPs rather than more general DKPs.
> 2. Agreed, we have rearranged Section 3.1 + 3.2 to start with a self-contained statement of the probability density of our generalised singular Wishart, and to push some material to the Appendix (also following R1's suggestion). We have also taken your suggestion with equation numbers and will only keep those that we reference.
> 3. DKPs aren't actually restricted to isotropic kernels - the requirement is that the kernel can be computed from the Gram matrix.  Indeed, the original DKP paper considered squared exponential and relu/arc-cosine kernels, which aren't isotropic.  We chose the squared exponential kernel because it is the standard choice in the DGP literature.  Indeed, while DGPs do have the flexibility to use a broader class of kernel, we do not know of work that actually uses this flexibility to improve performance on UCI.  If you can suggest such a kernel, we'd be happy to run those experiments.  Finally, it is worth noting that we can use any kernel at first layer of the DWP, where features (inputs) are available. We have clarified our choice of kernel in the main text.
> 4. This is a valid point: we have added a note about the restriction on kernel choice to the abstract (the modification is in our working manuscript, but we can't currently edit the abstract in OpenReview).

---

> > ### Comment · Area_Chair_Ar3c · 2021-08-22
> > **clarification**
> >
> > Hi authors, I have a question if I may:
> >
> > (I edited this comment after reading the paper)
> >
> > The reviewer seemed to be questioning whether there were any theoretical guarantees that suggest that DWPs should outperform DGP.  I agree with your point that capturing the symmetry allows a better approximation and so a better elbo and potentially a tighter PAC bound - but I think the question is really about whether this work is justified if there is no guarantee that DWPS will outperform DGPs. Perhaps you have some pointers?
> >
> > Are there some simple limiting cases where the DGP representation must win, or where the DKP representation must win?

---

> > > ### Author Response · Authors · 2021-08-23
> > > **Author response**
> > >
> > > Hi AC!  Thanks for your response!
> > >
> > > We can prove that the ELBO is lower if we are able to use "equivalent" approximate posteriors for the DGP and DWP.  In particular, we can write the DWP ELBO, $L_\text{DWP}$ and the DGP ELBO, $L_\text{DGP}$ as,
> > >
> > > $L_\text{DWP} = \log P(d) - D_\text{KL}(Q(G)|| P(G|d))$
> > >
> > > $L_\text{DGP} = \log P(d) - D_\text{KL}(Q(F)|| P(F|d))$
> > >
> > > where $d$ is the data, $F \in R^{P \times \nu}$ is the Gaussian process features and $G \in R^{P\times P}$ is the DWP Gram matrix, and we consider only a single-layer for simplicity.
> > >
> > > Critically, we know, that $F$ contains more degrees of freedom than $G$.  In particular, consider the low-rank case with the low-rank "Cholesky" $L(G) \in R^{P \times \nu}$ (see paper for details), such that $G = L(G) L(G)^T$, then, $F = L(G) U$ where $U \in R^{\nu \times \nu}$ is a unitary matrix.  Thus, a distribution over $F$ can be equivalently written as a distribution over $G, U$.  As KL-divergences are invariant to an invertible change of variables, we have,
> > >
> > > $L_\text{DGP} = \log P(d) - D_\text{KL}(Q(G) Q(U| G)|| P(G|d) P(U))$
> > >
> > > where for simplicity, we have used the fact that the posterior over $U$ is equal to the prior, as $U$ does not affect the predictions (see Aitchison et al. 2020, Appendix D2).  Separating out the terms relating to $G$ and $U$,
> > >
> > > $L_\text{DGP} = \log P(d) - D_\text{KL}(Q(G) || P(G|d)) - E_{Q(G)}[D_\text{KL}(Q(U| G) || P(U))]$,
> > >
> > > We note that the sum of the first two terms, $\log P(d)$ and $- D_\text{KL}(Q(G) || P(G|d))$ is equal to the DWP ELBO,
> > >
> > > $L_\text{DGP} = L_\text{DWP} - E_{Q(G)}[D_\text{KL}(Q(U| G) || P(U))]$
> > >
> > > And as the KL-divergence is non-negative, the DWP ELBO is always bigger than or equal to the DGP ELBO,
> > >
> > > $L_\text{DGP} \leq L_\text{DWP}$
> > >
> > > This intuition is one of the main reasons that function-space variational inference has become popular in recent years (see [1] for more theoretical results along these lines). However, in our case this bound is only valid where the DGP approximate posterior induces a distribution over $G$ that matches that in the DWP.  This would hold if we used multivariate normal posteriors for $F$ and non-central Wishart posteriors for $G$.  However, this isn't possible, because the non-central Wishart has an intractable probability density function (see Aitchison et al. 2020).  We are thus forced to use a different DWP approximate posterior over Gram matrices, so the exact relation doesn't necessarily hold. However, given sufficiently flexible approximate posterior, we should be able to see the advantages shown above, which we have demonstrated empirically (Table 1).
> > >
> > > Its this improved ELBO which should feed in to PAC Bayes to give better generalisation bounds.
> > >
> > > [1] https://arxiv.org/abs/2011.09421

---

### Official Review · Reviewer_ty8W · 2021-07-16

**Rating:** 7
**Confidence:** 4

**Summary:**

The authors introduce effective inference strategies for deep Wishart processes (DWP). This class of deep models is of interest because it moves away from the featurized representation implicit in most deep models to one that is fully kernelized. Inference relies on a doubly stochastic inducing-point scheme familiar from variational Deep Gaussian Processes and makes use of a custom generalization of the Bartlett decomposition adapted to the DWP setting in order to define a flexible variational family. In experiments on UCI data the method shows small but consistent improvements over Deep Gaussian Processes.

**Limitations And Societal Impact:**

I think it would be useful to better understand how the runtime compares e.g. to DGP inference. For example, if the scheme does in fact suffer from large gradient variance, then more optimization steps may be required, etc.

**Main Review:**

This is an interesting and technically sound submission. The clarity of exposition is generally high and the work is contextualized with respect to related lines of work in the literature. While the empirical benefits w.r.t. predictive performance of the proposed model and inference scheme are limited (when compared to Deep Gaussian Processes), what makes this approach particularly interesting is the fully kernelized approach that is taken. One might hope that further generalizations could pay larger predictive dividends. That the resulting scheme differs in a non-trivial way from more featurized representation commonly used in various approaches to deep Bayesian models can be read off, e.g., from the computational complexity, which does not depend on layer width (see Sec. 3.5).

More specific questions/comments:
- I'm confused about some of the notation. Should lowercase p be changed to uppercase P everywhere (e.g. Sec 3.1)? Is V_\ell defined correctly on line 163?
- Is Sec. 3.1 the best use of space? It might make sense to move some of these details to the supplementary material and instead focus on one or more of: i) unpack choices made in Eqn 28a; ii) make the comparison to DGP inference more detailed.
- Can you comment on the gradient variance of the sampling scheme in Sec. 3.4? In DGPs we can compute the distribution q(f) = \int du p(f|u)q(u) in closed form and sample from the marginals. This has the nice benefit that the different f_i are iid. Alternatively we could sample u and then sample each f_i conditioned on u, which results in larger gradient variance. If I understand it correctly, you are effectively in this latter regime and thus presumably experience somewhat large gradient variance. Is that correct? I guess this is more or less a direct consequence of maintaining a kernelized approach?
- You comment (line 230ff) that “to get optimization to work effectively Salimbeni & Deisenroth (2017) were forced to modify the prior to introduce skip connections." How do you know this is a problem with inference and not a problem with model specification? How can you disentangle the two? Shouldn't sufficiently deep DWPs exhibit some of the same pathologies explored in e.g. "Avoiding pathologies in very deep networks" by Duvenaud et. al.?

## After author response

I thank the authors for their response and I maintain my original score. While some of the concerns raised in the reviews have some merit (in particular w.r.t. the breadth of the experiments), I find some of the concerns raised by the more negative reviews less convincing (e.g. "might not attract broad interest"). While the topic explored in this submission is somewhat dry and technical, I believe that this contribution would make a valuable contribution to the conference. Formulating deep models in function space remains a somewhat underexplored (but potentially important) area, and for this reason (among others) I would argue that the solid technical contributions in this submission could be of interest to the NeurIPS community.

**Time Spent Reviewing:**

1.25

---

> ### Author Response · Authors · 2021-08-10
> **Response**
>
> Thank you for your review and helpful comments! To address your points:
>
> **Notation**: Lowercase $p$ should indeed be uppercase (including in the definition of $V_\ell$ on line 163); thank you for pointing this out!  Confusingly, $p_\ell$ is also a parameter of the approximate posterior.  We have changed the notation for $p_\ell$ to avoid this confusion, and have gone through carefully to make sure all the $p$s are uppercase. We have additionally done a further pass of the manuscript to ensure that the rest of the notation is consistent.
>
> **Sec 3.1**: We have rearranged Section 3.1 + 3.2 to start with a self-contained statement of the probability density of our generalised singular Wishart and to push some of the material in 3.1 to the Appendix. With the extra space, we have unpacked choices made in Eqn 28a and made the comparison to DGP inference more detailed.
>
> **Sampling us**: This is an interesting question! We agree that we are in essence sampling "u"s, and we haven't been able to find a way of integrating them out in the DWP framework.  That said, the impact on gradient variance relative to DGPs is unclear.  In general, analytically integrating out latent variables seems to reduce variance and improve learning in variational inference.  While we aren't able to integrate out the "u"s, by working with Gram matrices we in effect integrate out the "orientation" of the features, because the Gram matrices are invariant to unitary transformations of the features (Eq. 17). Thus, DGPs integrate over the u's  while DWPs integrate over the orientation of the features, and the question is: which gives a bigger reduction in gradient variance?  The only way of answering this question is empirically, but the approximate posteriors and parameterisations are so different that isn't clear a fair comparison is possible.
>
> **Runtime**: For example 5-layer DWPs take 0.368 seconds per epoch for boston, whereas the equivalent DGP takes 0.713 seconds per epoch. This is due to the lower computational cost for DWPs (see also response to R4). While fair comparisons are difficult without extensive hyperparameter optimization, we trained both DGPs and DWPs in 20,000 gradient steps, chosen because it is standard for DGPs on these datasets.  Given DWPs are faster per epoch, this corresponds to considerably less overall compute time for DWPs.  We will include plots of performance vs epoch and vs wallclock time in the final version.
>
> **Skip connections/identity mean function**: This is a good question. Salimbeni and Deisenroth (2018) suggest that they were inspired to introduce skip connections/an identity mean function because of issues with the prior discussed in Duvenaud et al. (2014; incidentally, this really is beautiful work, and it was a pleasure to look over it again!).  If the issue was predominantly with the prior, we would expect it to appear for any type of inference. Indeed standard DSVI seems not to train well on a network without skip connections (e.g. see Table 1 in the original DKP paper, for instance for boston). But we ran DGP baselines using global inducing methods from Ober and Aitchison (2020) which does seem to work well even without skip connections.  This leads us to suspect that the central issue is really inference, not the prior. In particular, the global inducing approximate posterior for DGPs and our DWP approximate posterior sample the features/Gram matrices conditioned on the previous layer features/Gram matrix.  As such, information propagates through the approximate posterior all the way from inputs to outputs and we get reasonable performance, even at initialization.  Interestingly, skip connections also allow information to propagate all the way from input to the outputs, and hence gives reasonable performance, even at initialization.  However, in a standard DGP without skip connections, the features at each layer are drawn from a posterior that factorises across layers (see Ober and Aitchison 2020 for details). As such, at initialization, the top-layer features have no dependence on the input at all (they just depend on how the approximate posterior was initialized).  Thus, at initialization, the performance of a standard DGP without skip connections can be very poor, and in certain situations, the DGP appears to get stuck close to that initialization and never get good performance.  That said, in _very_ deep networks, we indeed expect problems related to model specification described in Duvenaud et al. (2014) to arise similarly in DGPs and DWPs, as they define an equivalent prior over functions. We have changed the writing surrounding this issue to clarify our point.

---

> ### Comment · Area_Chair_Ar3c · 2021-08-22
> **post rebuttal update and discussion?**
>
> Hi and thanks for your review!
>
> Did the authors' rebuttal address your points well, are you standing by your score?

---

### Decision · Program_Chairs · 2021-09-27

**Decision:**

Accept (Poster)

**Comment:**

The authors propose a variational approximation scheme for Deep Kernel Processes (DKP). Building on previous work, the DKP is an alternative representation of the Deep Gaussian Process (DGP), where the representation is over Gramm matrices rather than over function evaluations.

Reviewers are in broad agreement that this work is a solid contribution in the development of DKPs, and in functional representation of Deep models in general. One reviewer raised a concern about the lack of theoretical guarantees that the new methodology brings over the DGP representation: in the rebuttal, the authors provide convincing arguments that the improvements in the ELBO due to the representation are directly connected to PAC bounds - I'd like to see a little discussion on this in the camera-ready version.